



# Evaluation of the coupled high-resolution atmospheric chemistry model system MECO(n) using in situ and MAX-DOAS NO$_2$ measurements

Vinod Kumar[1], Julia Remmers[1], Steffen Beirle[1], Joachim Fallmann[2], Astrid Kerkweg[3], Jos Lelieveld[1], Mariano Mertens[4], Andrea Pozzer[1], Benedikt Steil[1], and Thomas Wagner[1]

[1]Max Planck Institute for Chemistry, Mainz, Germany
[2]Institute of Meteorology and Climate Research, Karlsruhe Institute of Technology, Germany
[3]Institute of Energy and Climate Research 8, Troposphere, Forschungszentrum Jülich, Jülich, Germany
[4]Deutsches Zentrum für Luft- und Raumfahrt, Institut für Physik der Atmosphäre, Oberpfaffenhofen, Germany

**Correspondence:** Vinod Kumar (vinod.kumar@mpic.de)

**Abstract.** We present high spatial resolution (up to $2.2 \times 2.2$ km$^2$) simulations focussed over south-west Germany using the online coupled regional atmospheric chemistry model system MECO(n). Numerical simulation of nitrogen dioxide (NO$_2$) surface volume mixing ratios (VMR) are compared to in situ measurements from a network with 193 locations including background, traffic-adjacent and industrial stations to investigate the model's performance in simulating the spatial and temporal

variability of short-lived chemical species. We show that the use of a high-resolution and up-to-date emission inventory is crucial for reproducing the spatial variability, and resulted in good agreement with the measured VMRs at the background and industrial locations with an overall bias of less than 10%. We introduce a computationally efficient approach that simulates diurnal and daily variability in monthly resolved anthropogenic emissions to resolve the temporal variability of NO$_2$.

    MAX-DOAS measurements performed at Mainz (49.99 °N, 8.23 °E) were used to evaluate the simulated tropospheric

vertical column densities (VCD) of NO$_2$. We propose a consistent and robust approach to evaluate the vertical distribution of NO$_2$ in the boundary layer by comparing the individual differential slant column densities ($dSCD$s) at various elevation angles. This approach considers details of the spatial heterogeneity and sensitivity volume of the MAX-DOAS measurements while comparing the measured and simulated dSCDs. The effects of clouds on the agreement between MAX-DOAS measurements and simulations have also been investigated. For low elevation angles ($\leq 8°$), small biases in the range of -14 to +7% and

Pearson correlation coefficients in the range of 0.5 to 0.8 were achieved for different azimuth directions in the cloud-free cases indicating good model performance in the layers close to the surface. Accounting for diurnal and daily variability in the monthly resolved anthropogenic emissions was found to be crucial for the accurate representation of time series of measured NO$_2$ VMR and $dSCD$s and is particularly critical when the atmospheric lifetime of NO$_2$ is relatively long.

## 1 Introduction

Regional atmospheric chemistry and transport models are important for the study and forecasting of atmospheric processes at fine spatial resolutions. The high spatial resolution of these models allows us to resolve localized emissions (e.g., industrial



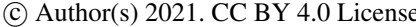



and urban clusters) and quantify their impacts on non-linear photochemical processes, e.g., ozone production (Vinken et al., 2014; Visser et al., 2019; Mertens et al., 2020). Various studies have shown that significant improvements in satellite retrievals can be achieved through the incorporation of highly resolved a priori trace gas and aerosol fields calculated by high-resolution regional models (Valin et al., 2011; Liu et al., 2020; Ialongo et al., 2020).

Regional models achieve high resolution by employing nesting around the location of interest, in which a fine-resolution model domain receives meteorological and chemical boundary conditions from a coarser resolution model spanning a broader area. The MECO(n) (MESSy-fied ECHAM and COSMO models nested n times) regional model system developed by Kerkweg and Jöckel (2012b) allows online coupling between different nests and in this way facilitates frequent updates of meteorological and chemical boundary conditions. The use of MESSy submodels for this coupled system also ensures consistent treatment of chemical speciation, chemistry and several other relevant processes governing the concentration of chemical species among various nests. A chemical evaluation of MECO(n) over Europe was performed by Mertens et al. (2016) for a set up at $\sim$ $12 \times 12$ km$^2$ spatial resolution for nitrogen dioxide (NO$_2$), ozone(O$_3$) and carbon monoxide (CO). The main strength of MECO(n), i.e., the online coupling with the COSMO model, makes it suitable for performing very high resolution (e.g. $< 3$ $\times$ $3$ km$^2$) simulations. For example, the operational COSMO model is already being used for weather forecasts at a spatial resolution of $2.8 \times 2.8$ km$^2$ by the German weather service (Deutscher Wetterdienst) and at $1 \times 1$ km$^2$ by the Federal Office of Meteorology and Climatology, Switzerland (MeteoSwiss). Similar high-resolution model simulations including chemistry have been shown to better represent local maxima (e.g., isolated point sources, road networks and ship tracks) and facilitate understanding of sector-specific impacts on secondary chemistry (e.g., ozone production) (Colette et al., 2014; Mertens et al., 2020). However, comparison with in situ measurements showed that these gains could only be quantitatively determined up to a resolution of $\sim 7 \times 7$ km$^2$, beyond which major improvements were not observed (Colette et al., 2014). In most of the cases, the improvement was limited by the resolution of the input emission inventory used in these studies, which are available at a much coarser resolution than that of the model set up. Apart from the coarse resolution, further limitations in such comparisons are imposed by the availability of mostly outdated anthropogenic emissions inventory and limited information about short-term temporal variability (e.g., day of the week or hour of the day) (Kuik et al., 2018). In most cases, input emission inventories are available at temporal resolutions of months to years, but in reality, emissions from several sectors (e.g., road transport, residential combustion) vary markedly depending on the hour of the day and day of the week. From a modelling perspective, however, incorporating high temporal resolution input emissions can be computationally inefficient due to the high readout time and subsequent requirement for interpolation on the model grid.

The evaluation of high-resolution mesoscale models is even more challenging due to the limited availability (in situ measurements) and unavailability (e.g. satellite observations) of reference datasets. The TROPOMI instrument aboard the Sentinel-5P satellite (Veefkind et al., 2012) has a high spatial resolution (up to $3.5 \times 5.5$ km$^2$) and is in principle well suited for comparison of the tropospheric vertical column densities (VCDs) simulated by the model. However, the limited temporal information – generally measurements from one overpass per day – precludes an evaluation of diurnal profiles, and hence TROPOMI is not well suited to demonstrating the advantages of considering diurnally varying input emissions. Besides VCDs, conventional model evaluation studies are often restricted to the evaluation of concentrations at discrete layers (most often at the surface)



using in situ measurements which are limited with respect to the temporal and spatial coverage. Evaluations of model vertical profiles are even rarer due to the paucity of vertically resolved measurements (e.g., balloon-based and aircraft measurements), which are technically very challenging. For example, discrepancies between regional models (WRF-Chem and CHIMERE)

and measurements were proposed to arise due to inappropriate parametrization of the turbulent diffusion constant; however, this could not be verified due to the lack of vertically resolved $NO_2$ measurements (Kuik et al., 2018; Schaap et al., 2015). Mertens et al. (2016) evaluated the vertical profiles of $O_3$ simulated by MECO(n) using ozone sonde data, but this provides little information about the variability within the boundary layer which is masked by that in the upper troposphere (above 400 hPa or $\approx 7$ km).

In this regard, MAX-DOAS measurements (Hönninger et al., 2004) provide a unique opportunity for model evaluation for a larger representative area (a few square kilometers), and over long temporal scales. To our knowledge, regional model comparison studies with MAX-DOAS are very limited (e.g. Shaiganfar et al. (2015); Vlemmix et al. (2015); Blechschmidt et al. (2020)), and mostly focus on the tropospheric VCDs. These studies have shown moderate correlations between MAX-DOAS and regional model calculated VCDs with major differences arising due to inappropriate representation of anthropogenic

emissions, differences between model-simulated and actual wind vectors, the presence of clouds and uncertainties related to the MAX-DOAS VCD retrieval. Further limitations in these comparisons arise due to assumptions of horizontal homogeneity and the challenges associated with accurately defining the area/volume for which the MAX-DOAS measurements are sensitive. In addition to the VCDs, there is additional information content in the MAX-DOAS measurements (e.g. spatial distribution of trace gases), which comes from scans performed at different elevation angles, thus probing the atmosphere along different

light paths. This information can be utilized to evaluate the performance of regional models in accurately simulating the spatial distribution within the boundary layer.

In this paper, we present high-resolution up to $(2.2 \times 2.2$ km$^2)$ MECO(3) simulations of $NO_2$ over south-west Germany using a high spatial resolution and up to date input emission inventory. We also effectively account for the day-of-the-week and diurnal variability in anthropogenic emissions in the model simulations by applying sector-specific hourly scaling factors

to the monthly-resolved anthropogenic emissions. The model description, details of MAX-DOAS measurements and analyses and other reference data are provided in section 2. We evaluate the model performance with respect to two input emission inventories and temporal resolution of emissions through comparison with in situ measurements in section 3.2 and MAX-DOAS tropospheric $NO_2$ VCDs in section 3.3. The TROPOMI comparison will be the focus of a future study. In this paper, we introduce a sophisticated and consistent approach for MAX-DOAS comparison, which overcomes the limitations of previous

such comparisons. We also evaluate the performance of the model in reproducing the vertical distribution of $NO_2$ within the boundary layer.



## 2   Methods

### 2.1   Model description

We use the one way coupled model system MECO(3) (Kerkweg and Jöckel, 2012a, b) based on MESSy version 2.54 (Jöckel
et al., 2010), which couples the global chemistry climate model EMAC (Jöckel et al., 2006) one way to the regional model
COSMO-CLM/MESSy (called COSMO/MESSy hereafter). COSMO-CLM is the community model of the German regional
climate research community jointly further developed by the CLM-Community (Rockel et al., 2008). The core driving model
for EMAC is ECHAM5 version 5.3.02 (Roeckner et al., 2003) and for COSMO/MESSy it is COSMO 5.00_clm10 (Rockel
et al., 2008; Steppeler et al., 2003). EMAC is configured with T106L31ECMWF spectral resolution corresponding to a grid
resolution of $\sim 1.1° \times \sim 1.1°$, extending up to 10 hPa ($\approx$30 km over Europe) vertically in 31 vertical layers and time step of
360 s. The global model meteorology (temperature, vorticity, surface pressure and divergence) is nudged to 6 hourly ECMWF
ERA-Interim reanalysis data. Model simulation are performed for May 2018, i.e., from `01-05-2018 00:00:00` until
`01-06-2018 00:00:00`. The initial chemical conditions for the regional model instances are provided by an EMAC sim-
ulation starting three years before the MECO(n) simulations start, i.e. on `01-03-2015`.

The three instances of the COSMO/MESSy model are coupled online one way from coarser resolution to finer resolution.
The first instance of COSMO/MESSy has a spatial resolution of $0.44° \times 0.44°$ ($\sim 50 \times 50$ km$^2$; referred to as CM50 hereafter),
the second one at $0.0625° \times 0.0625°$ ($\sim 7 \times 7$ km$^2$; referred to as CM07 hereafter) and the third one at $0.02° \times 0.02°$ ($\sim 2.2$
$\times 2.2$ km$^2$; referred to as CM02 hereafter). All three COSMO/MESSy domains are set up in a rotated coordinate system with
the location of the north pole at 40 °N and -170 °E. The CM07 domain is focussed around Germany, and CM02 further zooms
in to south-west Germany, as shown in Figure 1. CM50 and CM07 have 40 terrain-following vertical levels extending up to
22.7 km while CM02 has 50 terrain-following vertical levels extending up to 22 km. In all the three domains, the lowermost
atmospheric model layer has a thickness of 20 m, while the lowest 1 km is split into 11 levels for CM50 and CM07 and 12
levels for CM02. The thickness of the vertical layers increases with altitude. The time steps for CM50, CM07 and CM02 are
s, 60 s and 20 s, respectively. The online coupling enables the specification of boundary conditions at each time step by the
respective driving model and is particularly advantageous for complex atmospheric chemistry modelling involving hundreds of
chemical tracers. Convection is parametrized according to the Tiedtke-Bechtold scheme for both CM50 and CM07 domains,
while for the CM02 domain, only shallow convection is parametrized according to the Tiedtke scheme (Bechtold et al., 2001;
Tiedtke, 1989).

The MECO(n) model set up achieves a very high consistency within the model chain, as all four model instances (EMAC
and three times COSMO/MESSy) use MESSy and thus imply the very same chemical speciation and process formulations
for the chemical processes (e.g., online/offline emission of chemical tracers (ONEMIS/OFFEMIS), chemistry (MECCA), dry
deposition (DDEP), wet scavenging (SCAV) and photolysis (JVAL)) (Jöckel et al., 2010; Tost et al., 2006; Kerkweg et al.,
2006a; Sander et al., 2014; Kerkweg et al., 2006b). Since deep convection is resolved in the CM02 domain, the MESSy
submodel CVTRANS (Tost et al., 2010) used to calculate the tracer transport due to convection is not used in this domain.
Consequently, convective scavenging and convective rain flux are also disabled in the CM02 domain when using the SCAV





submodel (Tost et al., 2006). For the chemical mechanism in MECCA, we use the Mainz Isoprene Mechanism (MIM 1) (Pöschl et al., 2000), including 142 gaseous chemical species involved in 236 gas and multiphase reactions and 74 photochemical reactions.

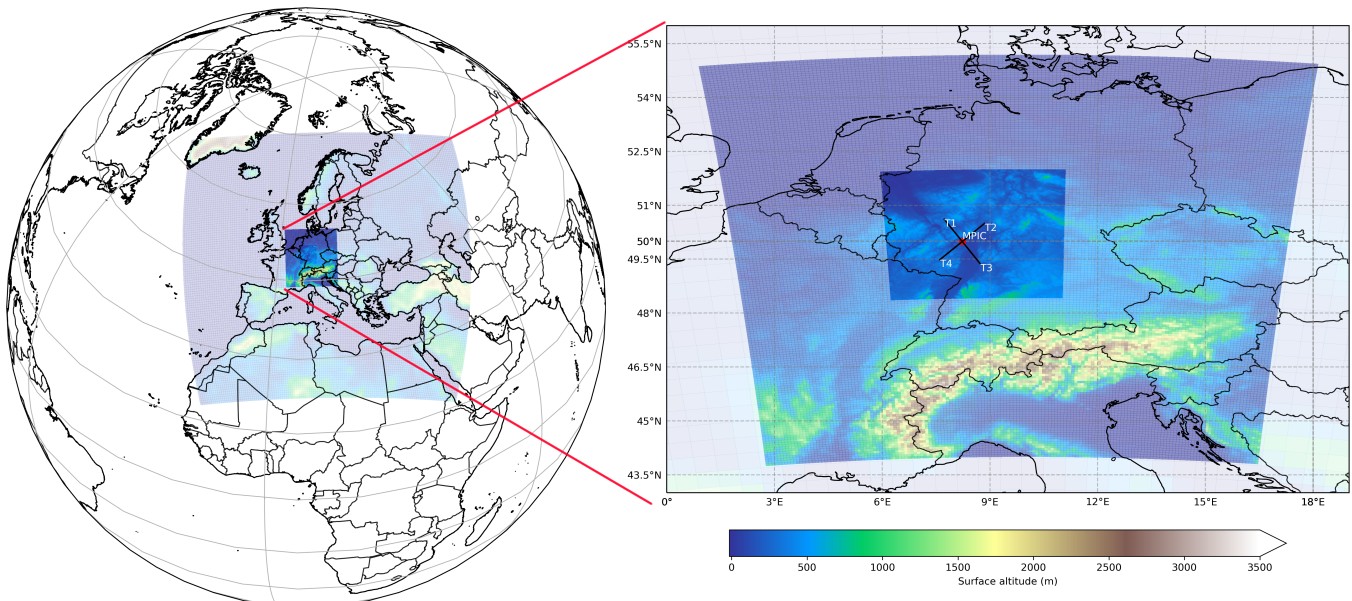

**Figure 1.** MECO(3) domains in the left panel colour-coded according to the surface altitude. A close up of the map in the right panel shows the CM07 and CM02 domains. The location of MPIC is shown as the red dot in the right panel with the arrows pointing in the viewing direction of the four telescopes.

For the global model and the CM50 domain, EDGAR 4.3.2 anthropogenic emissions ($0.1° \times 0.1°$ globally) (Crippa et al., 2018) have been used, while for the CM07 and CM02 domain, two different anthropogenic emission scenarios, namely TNO MACC III (Kuenen et al., 2014) (available for Europe, until 2011) and UBA (Strogies et al., 2020) (Umweltbundesamt; available for Germany, until 2018) have been employed. Total annual anthropogenic emissions of $NO_x$ within Germany are 366 Gg(N) and 214 Gg(N) for UBA (for 2018) and TNO MACC III (for 2011), respectively. For the TNO MACC III and UBA emissions, $NO_x$ was originally expressed as $Kg\,year^{-1}$ and $Kt\,year^{-1}$, respectively, of $NO_2$, which were further converted as $molecules\,m^{-2}\,s^{-1}$ for use in model simulations. Three MECO(3) simulations were performed with differing fine temporal variation for the TNO MACC III and UBA anthropogenic emissions, as summarized in Table 1.

The subscript '$di$' in Table 1 indicates the use of diurnal and day-of-the-week variability in $NO_x$ and CO emissions from the road transport and residential and non-industrial combustion sectors (see appendix A for further details). For the $TNO_{fl}$ and $UBA_{fl}$ set ups, constant anthropogenic emissions are used for the complete month. The sector-wise anthropogenic emissions are imported via the IMPORT submodel (Kerkweg and Jöckel, 2015). For specifying the temporal profiles (diurnal and day-of-the-week) in the anthropogenic emissions, we first created hourly resolved time series of scaling factors to be applied to



**Table 1.** Model set ups

| Simulation id | Anthropogenic emissions (in CM07 and CM02 domains) | Temporal resolution of emissions |
|---|---|---|
| $TNO_{fl}$ | TNO MACC III | Monthly |
| $UBA_{fl}$ | UBA (for Germany) and TNO MACC III (outside Germany) | Monthly |
| $UBA_{di}$ | UBA (for Germany) and TNO MACC III (outside Germany) | Hourly |

the monthly mean values using the factors shown in Figure A1. Please note that the factors have a weekly cycle, and these are normalized such that the total emission over a week is conserved for a given sector. Individual hourly time series of emission scaling factors are imported via IMPORT_TS (Kerkweg and Jöckel, 2015). The scaling factor for a specific model time step

is calculated by interpolating the time series and applying to the monthly emissions and subsequently, the emissions flux and tendency (change in VMR per model time step) are calculated using the ONEMIS submodel (Kerkweg et al., 2006b). The MESSy OFFEMIS submodel (Kerkweg et al., 2006b) updates the tendencies for emissions from the sectors for which a finer temporal profile is not necessary (e.g., agriculture, waste management, refineries).

The anthropogenic emissions are vertically distributed depending on the source sectors according to the recommendation by

Pozzer et al. (2009) except that the lowest injection height is reduced to 10 m as opposed to 45 m. This was necessary because the lowest level of COSMO extends from the surface to 20 m altitude, while the median lowest level height of EMAC (as used by Pozzer et al. (2009)), is about 60 m. Lightning $NO_x$ is calculated for the global model according to the parametrizations by Grewe (2009) and transferred to the subsequent instances of COSMO using the Multi-Model-Driver (MMD) coupling of MMD2WAY submodel (Kerkweg et al., 2018). The lightning frequency was scaled to produce 2.5 $Tg(N)$ $year^{-1}$ globally. Soil

$NO_x$ and biogenic emissions (e.g., isoprene and monoterpenes) are calculated online using the ONEMIS submodel separately in EMAC and individual COSMO/MESSy instances. For May 2018, soil $NO_x$ emissions were calculated to be 17.8 and 2.4 $Gg(N)$ for the CM07 and CM02 domains, respectively. Soil $NO_x$ emission over entire Germany in the CM07 domain was 5.9 $Gg(N)$ for May 2018. Non-methane volatile organic compounds (NMVOCs) emissions were also provided as lumped group of species, which were speciated according to the recommendation by Huang et al. (2017).

## 2.2   4-Azimuth MAX-DOAS measurements

Multiple AXis Differential Optical Absorption Spectroscopy (MAX-DOAS) measurements were performed using a custom-built instrument installed at the rooftop of the Max Planck Institute for Chemistry (MPIC) building (49.99 °N, 8.23 °E, 150 m amsl; the red dot in Figure 1). The instrument consists of 4 telescopes (T1, T2, T3 and T4 pointing at azimuth angles of 321, 51, 141, and 231°, respectively clockwise from the north). The intersecting arrows in Figure 1 indicate the azimuth direction of the

four telescopes. Individual optical fibre bundles transmit the light from the respective telescopes to a temperature-controlled spectrograph. The spectrograph consists of a 2-dimensional (1023 × 255) CCD detector array. The incoming light of the four telescopes is projected to different row regimes of the CCD. This set up reduces the instrumental differences between the measurements to a minimum. The measurements are performed along the four viewing directions simultaneously such that all





the telescopes (T1-T4) point towards the same elevation angle (EA). One complete measurement sequence for each telescope
involved measurements at eight off-axis elevation angles (1, 2, 3, 5, 8, 10, 15 and 30°) and in the direction of the zenith.
The field of view at 1° elevation angle was blocked partially for the different telescopes and hence was discarded from the
subsequent analyses. We applied the DOAS principle (Platt and Stutz, 2008) to the measured spectra to retrieve the elevation
angle dependent differential slant column densities ($dSCD$) of $NO_2$ and the oxygen dimer ($O_2$-$O_2$ or $O_4$) adapting to the fit
setting described in Table C1. The $dSCD$s can be regarded as the difference between the concentration integrated along the
light path at a chosen elevation angle and the concentration integrated along the direction of the zenith. This approach is used
to eliminate the stratospheric information and retrieve the tropospheric contribution.

In order to retain only the highest quality DOAS fit results, we discarded all retrievals with fit rms (root mean square) values
greater than $1.0 \times 10^{-3}$. $NO_2$ VCDs are retrieved using the geometric approximation on the measured $dSCD$s at 30° elevation
angle. Since MECO(3) was configured to write the output for the CM02 domain at an hourly frequency as mean values, we
also average the MAX-DOAS retrieved quantities (see section 2.2.2) at a similar frequency while discarding retrieval with high
spectral analysis rms values.

### 2.2.1 Cloud classification

A cloud classification was performed using MAX-DOAS measurements of the colour index (CI; the ratio of measured signal at
330 nm and 390 nm) and the $O_4$ $dSCD$s according to the method described by Wagner et al. (2016). In order to generate robust
thresholds for the cloud classification, one month of data is not sufficient, and hence we used a longer time series of MAX-
DOAS measurements from 27.03.2018 until 14.09.2018. We performed cloud classification separately using measurements
performed by the four telescopes. Figure C1 summarizes the cloud conditions for all the days of May 2018 for telescope T2.
Briefly, clear sky conditions were observed from `05-05-2018` until the afternoon of `09-05-2018`. On other days, cloudy
conditions were observed for several hours with sky conditions alternating between broken clouds, continuous clouds and
optically thick clouds. The cloud classification results for the other telescopes were similar to that of T2.

### 2.2.2 Retrieval of differential box air mass factors using 3D aerosol profile inversion

As mentioned above, the $dSCD$s retrieved from MAX-DOAS measurements depend on the differential light path between the
off-axis (EA = $\alpha$) and zenith measurements. $dSCD$s are related to the VCDs via the differential air mass factors ($dAMF$s)
according to the following equation:

$$VCD = \frac{dSCD_\alpha}{dAMF_\alpha} \tag{1}$$

The $O_4$ mixing ratio is almost constant throughout the troposphere and its VCD only depends on the atmospheric temperature
and pressure profile. Hence, using measured $O_4$ $dSCD$s and the knowledge of $O_4$ VCDs, the corresponding $dAMF$s can be
calculated. If we visualize the atmosphere in several discrete layers, the partial $dSCD$ in a specific layer ($k$) would be related
to the partial VCD ($V_k$) and the differential box air mass factor ($dbAMF_{\alpha,k}$) would be specific for the layer $k$ in a similar way





as in equation 1. $dAMF$ can be reconstructed from the $dbAMF_{\alpha,k}$ according to equation 2:

$$dAMF_\alpha = \frac{\sum_k V_k \times dbAMF_{\alpha,k}}{\sum_k V_k} \qquad (2)$$

The presence of aerosols can change the light path, and hence the $dSCD$s (and consequently the $dAMF$s). Profile inversion algorithms can find the optimal aerosol extinction profiles corresponding to the measured $O_4$ $dSCD$s for a sequence of elevation angles (Wagner et al., 2004; Clémer et al., 2010; Wagner et al., 2011). This can be subsequently used to calculate
the $dbAMF_{\alpha,k}$ in the discrete atmospheric layer indexed by $k$. In addition to the $O_4$ VCDs and measured $dSCD$s, the profile inversion algorithms require an offline look-up table of $O_4$ $dAMF$s corresponding to various combinations of measurement geometry and aerosol extinction profiles calculated using radiative transfer models (e.g., McArtim; Deutschmann et al. (2011)).

We used the profile inversion algorithm $\pi$-MAX (Parameterized profile Inversion for MAX-DOAS measurements) (Remmers et al., in preparation), for the retrieval of the $dbAMF$s. In comparison to the traditional parametrized profile inversion
algorithms (e.g., MAPA; (Beirle et al., 2019)), which only parametrizes the aerosol optical depth (AOD) and vertical profiles of aerosol extinction (e.g. shape ($s$) and height ($h$) of the profile) for a 1D retrieval (along altitude), $\pi$-MAX includes additional parameters related to the horizontal gradients in the viewing direction. Figure 2 shows the schematic of a traditional profile

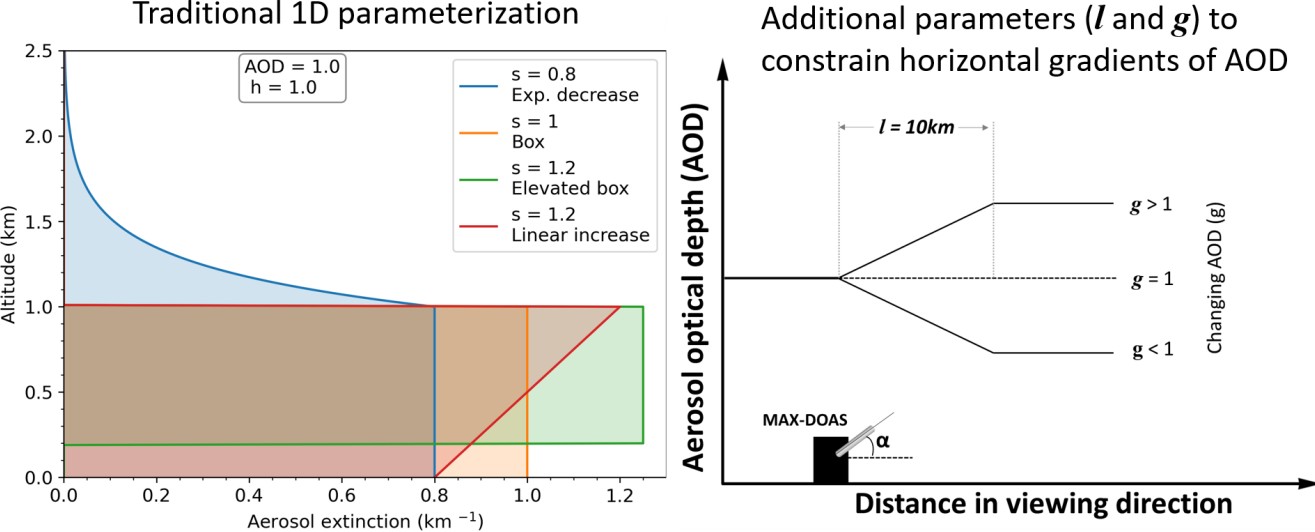

**Figure 2.** Left) Schematic illustration of traditional 1D parametrized profile inversion which constrains AOD, height (h) and shape (s) of the profile. The example shown here correspond to a scenario with AOD = 1.0, h = 1.0 and four different profile shapes representing exponential decrease, box profile, elevated box profile and linearly increasing profiles. Right) Additional parameters $l$ and $g$ further constrain the horizontal gradients of AOD for 2D profile inversion. Here, $g$ denotes a linear change in AOD for a distance $l$ along the viewing direction of the MAX-DOAS instrument.

inversion for an example case of AOD = 1.0, h = 1 and various parametrizations of $s$ representing the respective profile shapes as well as additional parameters $g$ and $l$ for $\pi$-MAX. These additional parameters describe the linear aerosol extinction change



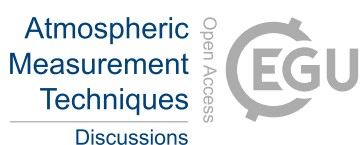

($g$) from the telescope location to a specific distance ($l$). Hence, they allow retrievals of 2D $dbAMF$s (and aerosol extinction profiles) as a function of distance from the telescope and altitude from the instrument, if measurement in only one azimuth direction is considered. If measurements in several azimuth directions are combined, 3D retrievals can be performed. In the current $\pi$-MAX set up, $l$ is fixed to 10 km. The $dSCD$ measurements in all four directions are used simultaneously with the constraint that the profile at the origin (location of the instrument) is the same for all the telescopes.

The quality of the profile retrieval from $\pi$-MAX can be qualified using the rms of the $dSCD$ fit corresponding to each complete elevation sequence. In order to retain only the highest quality profile inversion results, we have retained retrievals corresponding to rms values less than 0.04 times the $O_4$ VCDs.

### 2.3 In situ chemical and meteorological measurement data

We used the surface temperature, relative humidity and wind measurement data from the climate data centre of the German
Weather Service (Deutscher Wetterdienst) for meteorological evaluation in the CM02 set up (DWD, 2019). Hourly measurements of surface temperature, relative humidity and wind speed are available for 304, 501 and 283 stations, respectively, in Germany for May 2018; out of these 178, 197 and 95, respectively, fall within the CM02 domain.

In situ measurements of $NO_2$ and $O_3$ are available from the German Environment Agency (Umweltbundesamt) (Minkos et al., 2019) from 410 and 266 stations, respectively, across Germany for May 2018. Among the 410 $NO_2$ measurement sta-
tions, 193 fall within the CM02 domain, out of which 119, 60 and 14 stations represent background, traffic and industrial locations, respectively. For $O_3$, 120 stations are within the CM02 domain, out of which 109, 3 and 8 stations represent the background, traffic and industrial locations, respectively. For most of the stations within the CM02 domain, $NO_2$ is measured online using the chemiluminescence method, in which $NO_2$ is reduced to NO using a heated molybdenum converter prior to its detection (Eickelpasch and Eickelpasch, 2004). Only at Schmücke (DEUB029), a photolytic converter is used in place of
the molybdenum converter, whereas at Pfälzerwald-Hortenkopf (DERP017), a CAPS (Cavity Attenuated Phase Shift Spectroscopy) instrument is used for measurement of $NO_2$ (https://www.env-it.de/stationen/public/downloadRequest.do). $O_3$ is measured online using UV absorption technique at all the stations. The measured in situ data are available at 1-hour resolution.

## 3 Results and Discussion

### 3.1 Meteorological evaluation: surface temperature, relative humidity and wind speed

In the online coupled MECO(n) system, COSMO/MESSy instances are not nudged directly towards the reanalysis dataset. Rather, these receive the meteorological boundary conditions from EMAC for the first instance and from the antecedent COSMO/MESSy for each subsequent instance on the four sides of the domain and the damping layer (ca. 11 km for CM50 and CM07 and 10.7 km for the CM02 domain). Hence individual COSMO/MESSy instances of MECO(n) can develop their own dynamics, which might result in a deviation from the actual meteorology. Hofmann et al. (2012) have evaluated the MECO(n)
meteorology and demonstrated comparable performance with respect to a similar model with offline coupling. Here, we briefly



evaluate the performance of MECO(n) in the CM02 set up with respect to the measured surface temperature, relative humidity and wind speed close to the surface.

The ability of the model to reproduce the temporal variability at multiple measurement stations can be evaluated using Taylor diagrams (Taylor, 2001), where we show the Pearson correlation coefficient ($R$), relative root mean square difference (RMSD) and relative standard deviation (RSD) with respect to the hourly resolution measured data. The Taylor diagrams for these parameters are shown in Figure 3.

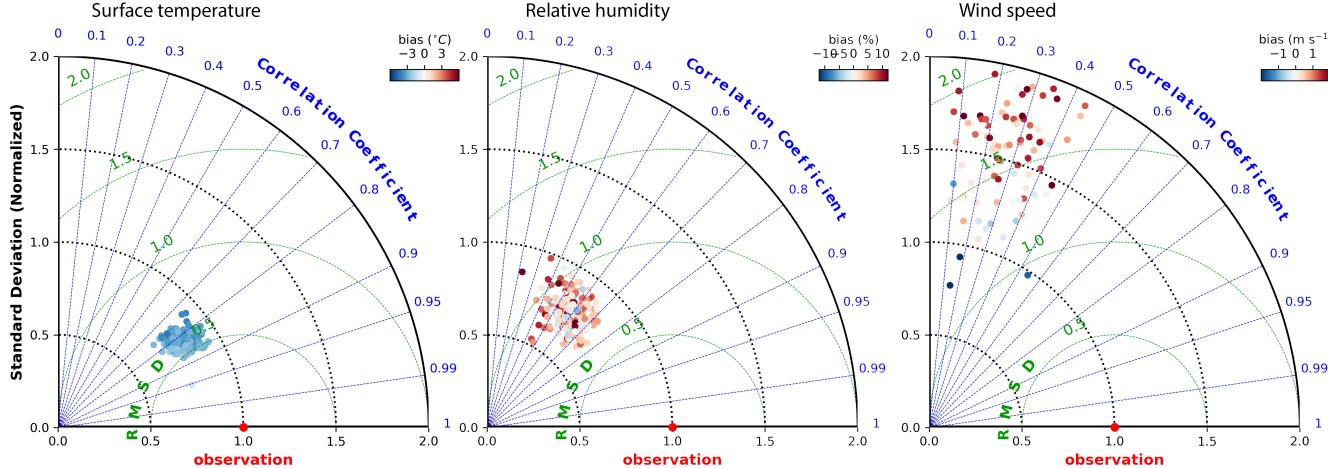

**Figure 3.** Taylor diagrams showing the agreement between measured and model-calculated surface temperatures (left) and relative humidity (centre) and wind speed (right). Each data point corresponds to an individual measurement station, with a total of 178, 197 and 95 stations for surface temperature, relative humidity, and wind speed, respectively

For surface temperature, the trends in the hourly resolved time series agree quite well with Pearson correlation coefficients generally between 0.8 and 0.9. The spatial patterns of the surface temperatures are also represented very well as inferred from small and precise RMSD values of circa 0.5. RSD values of less than 1 indicate that observed temporal variability in the model has a smaller amplitude than that of the measurements. There is a cold bias of ∼3 °C across the domain, which is similar to that observed by Mertens et al. (2016) for Germany in summer. Previous long term evaluation of the COSMO-CLM model has shown a cold bias of 2 – 2.5 °C compared to observation of the annual mean surface temperature over Germany, which increases in the summertime (Böhm et al., 2006). This bias is most probably due to inaccurate representation of root depth and soil temperature damping in the soil model. For relative humidity, the trends in the hourly resolved time series agree reasonably well with Pearson correlation coefficient generally between 0.5 and 0.7. Both positive and negative mean biases are observed for the different stations. For wind speeds, the Pearson correlation coefficients are generally between 0.2 and 0.5, but the bias was generally small and in the range of $\pm 1\,\mathrm{m\,s^{-1}}$.





## 3.2 Evaluation of surface mixing ratios of $NO_2$

In this section, we present the model results for simulated $NO_2$ surface volume mixing ratios (VMR) and compare with the

in situ observations for May 2018. Figure 4 shows the spatial distribution of monthly mean $NO_2$ VMRs in the lowest vertical layer (0-20 m) for the CM02 domain for the three model set ups listed in Table 1. The monthly mean VMRs from the in situ measurement stations are depicted as square, circle and pentagon markers for background, traffic-adjacent and industrial sites, respectively, overlaid on the maps using the same colour scale as that for simulated VMRs.

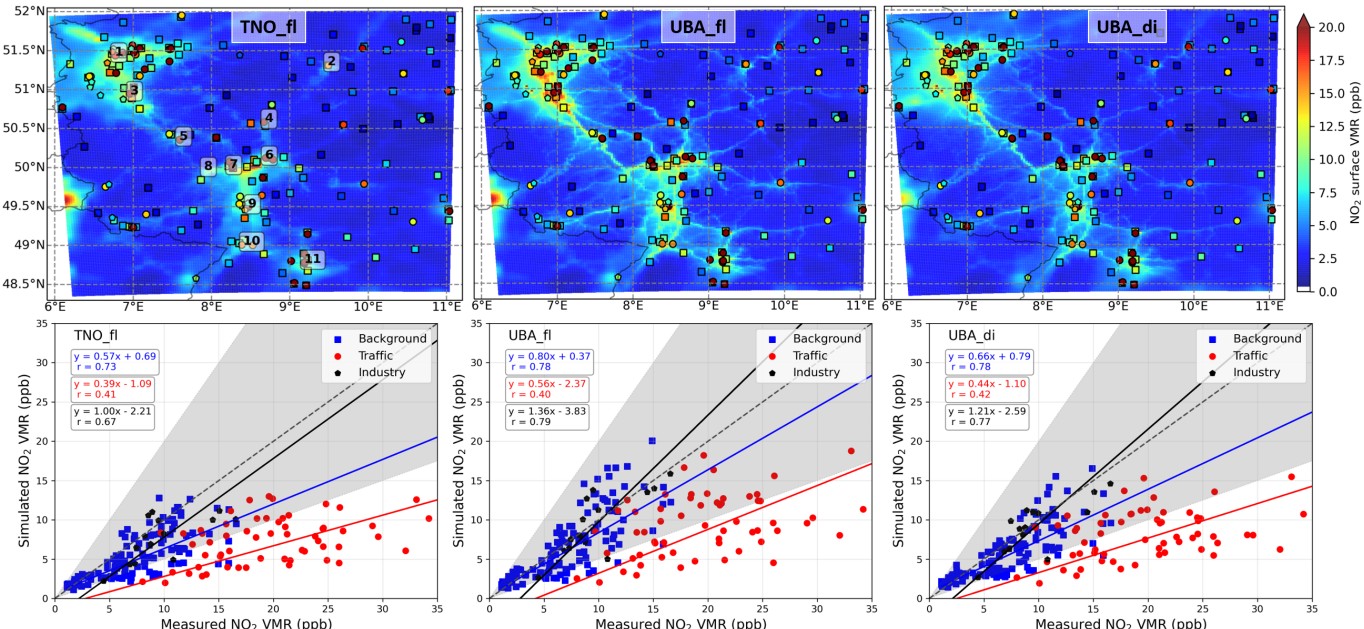

**Figure 4.** Top) Spatial distribution of monthly mean $NO_2$ surface VMRs for the three simulations using different emission inventories (left: TNO, middle: UBA without diurnal variations, right: UBA with diurnal variations) for CM02 for May 2018. The square, circle and pentagon markers overlaid on the maps represent the monthly mean measured $NO_2$ VMRs for background, traffic and industrial locations, respectively. The numbers on the top left panel correspond to the following German cities in decreasing order of latitude coordinate values, i.e., 1: Duisburg, 2: Kassel, 3: Köln, 4: Gießen, 5: Koblenz, 6: Frankfurt, 7: Mainz, 8: Bingen, 9: Mannheim, 10: Karlsruhe 11: Stuttgart. Bottom) Scatter plot and orthogonal distance regression weighted by the inverse of the square of the standard deviation of simulated monthly mean $NO_2$ surface VMRs with respect to the in situ measured values for different station types (background: blue square; traffic: red circle, and industrial: black pentagon).

Overall, the spatial distribution of $NO_2$ VMRs is as expected, such that the high values are observed in densely populated

areas, e.g., the Ruhr area, Luxembourg, around Frankfurt, Mannheim, Karlsruhe and Stuttgart. For the simulation with the high-resolution UBA emissions, we observe many details in the $NO_2$ surface concentration with higher values coinciding with the major motorways of Germany which were not so obvious with $TNO_{fl}$ (e.g., A61 motorway between Köln and Bingen,





A3 between Frankfurt and Bingen, A48 and A1 connecting Koblenz and Luxembourg and A4 and A9 between Gießen and
Leipzig). The performance of the model in reproducing the spatial variability can be quantitatively described using the root
mean square deviation (RMSD) between the monthly mean measured and simulated NO$_2$ VMRs for all the measurement
stations combined. We note that using the high-resolution UBA emissions improves the RMSD for background locations from
3.3 ppb ($\sim 45\%$ of the measured mean) for TNO$_{fl}$ to 2.7 ppb ($\sim 37\%$). Since UBA emissions are up to date and are available
for the same year as that of simulation, the mean bias for the background locations also improves from -2.0 ppb (-27%) for
TNO$_{fl}$ to -0.5 ppb (-7%) for UBA$_{fl}$. At locations near heavy traffic, the bias improved from -12.5 ppb (-63%) for TNO$_{fl}$ to
-10.4 ppb (-52%) for UBA$_{fl}$.

Even though the anthropogenic NO$_x$ emissions have reduced by $\sim 15\%$ over Europe from 2011 (the most recent year for
which TNO emissions are available) to 2017 (EEA, 2019), total NO$_x$ emissions over Germany are $\sim 40\%$ lower in TNO
MACC III as compared to UBA. For TNO MACC III, NO$_x$ emissions are lower across all the sectors except for ship transport.
For the two strongest NO$_x$ emission sectors, e.g. road transport and energy industries, TNO MACC III NO$_x$ emissions (for
2011) are $\sim 30\%$ and 58%, respectively, lower than the UBA NO$_x$ emissions (for 2018). Recent top-down emissions estimates
over urban centres in Germany have also pointed towards an underestimation of as much as a factor of 2 from the transport
sector and 1.5 overall (Kuik et al., 2018) by the TNO MACC III emissions, even by considering a conservative approach. The
underestimation of the a priori NO$_x$ emissions is the most important factor for the large negative bias in the TNO$_{fl}$ set up.
Getting up to date emission inventories is difficult, and we could only get this data for Germany, but for future studies with
simulation involving larger domains which include more countries, it is recommended to use more current emission inventories.

Unlike for the background locations, we do not observe a major improvement for the traffic-adjacent locations using the
UBA emissions. Even at such high spatial resolution ($2.2 \times 2.2$ km$^2$), the spatial smoothing leads to insufficient reproduction
of peaks for locations close to strong emission sources as also documented by Shaiganfar et al. (2015). Another factor which
could contribute to the differences between measured and simulated NO$_2$ is related to the chemiluminescence principle used
for measurements: NO$_2$ is first reduced to NO before subsequently reacting with O$_3$ generated within the analyser. This is
known to overestimate the actual NO$_2$, because the molybdenum converter within the analyser also reduces the NO$_x$ reservoir
species (e.g., HNO$_3$, PAN) to NO prior to detection (Dunlea et al., 2007). PAN and HNO$_3$ are more abundant at the traffic
and urban locations with a combined monthly mean mixing ratio of between 0.9 and 1.1 ppb in the UBA$_{di}$ set up. This could
account for 3 – 10% of the measured NO$_2$ at the traffic-adjacent locations.

Figure 5 shows a comparison of measured and the simulated NO$_2$ surface VMRs in the CM02 domain as Taylor diagrams
for the three different simulations. Both for TNO$_{fl}$ and UBA$_{fl}$, we observed rather poor agreement of the hourly temporal
variations of the measured VMRs with several stations even showing negative values of $R$. In Germany, transport emissions
account for $> 45\%$ of the total NO$_x$ emissions, which show a large diurnal variability (greater than 200 % peak to peak; see
Figure A1). These variabilities are generally not taken into account for regional model simulations and have shown to cause
larger bias during peak traffic hours on weekdays (Kuik et al., 2018). In the UBA$_{di}$ set up, accounting for diurnal and day-
of-the-week variability in the anthropogenic emissions shows significant improvement with $R$ values of between 0.3 and 0.6,
smaller RMSD values and more consistent agreement for different stations. However, we also note that overall negative bias is





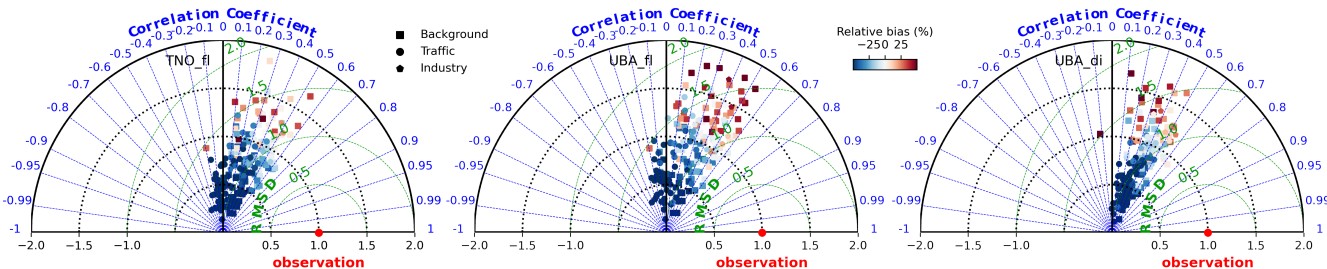

**Figure 5.** Taylor diagrams showing the Pearson correlation coefficients, normalized standard deviation and normalized root mean square difference corresponding to hourly resolution measured and simulated $NO_2$ VMRs (in CM02 domain) for background, traffic, and industrial sites represented as square, circle and pentagon markers, respectively. The left, middle and right panels correspond to the $TNO_{fl}$, $UBA_{fl}$ and $UBA_{di}$ set ups, respectively.

increased for the $UBA_{di}$ set up compared to $UBA_{fl}$. The diurnal profiles of emissions in the transport sector increase the $NO_x$ amount to more than twice as much during the daytime when the atmospheric lifetime is lower and decreases to less than one

quarter during the night when lifetime is high. Hence, overall, the monthly mean surface $NO_2$ VMR decreases when a diurnal profile is applied to $NO_x$ emissions. The normalized standard deviation (relative to the standard deviation of measured VMRs) improves by inclusion of diurnal profiles, as we observe more stations with values in the $0.5 - 1.5$ range.

Using the different COSMO/MESSy instances of the MECO(3) system, we are able to investigate the importance of model resolution if the same emission inventory is used for different model resolutions. Figure C2 shows the spatial distribution of

simulated $NO_2$ surface VMRs and the agreement with measurements for the CM07 set up in a similar way as Figure 4 for CM02. We note that for $TNO_{fl}$, there is only very little further detail in the spatial distribution for CM02 as compared to CM07. Meteorology (e.g. wind patterns) might be better resolved using a fine model resolution on individual days. Still, when averaged over several days, these will be smoothed, and the spatial patterns would be limited by the resolution of the input emissions inventory. Consequently, we also did not observe any significant improvement in the RMSD (45%) between the

monthly mean measured and simulated $NO_2$ VMRs as compared to the CM07 simulation (RMSD = 48%) using the TNO MACC III emission inventory. In contrast to this, using the high-resolution UBA emissions, the spatial details as depicted in CM02 smear out in the CM07 instance. For both, $UBA_{fl}$ and $UBA_{di}$ set ups, the RMSD improves from $\sim 45\%$ for the CM07 to $\sim 37\%$ for CM02, showing the added value of the higher resolution simulation. Hence, for studies where small scale variability is crucial, it is important to use a high-resolution model together with an input emission inventory of similar resolution.

Further reasons which could account for the lower bias of the $NO_2$ VMR in the model could be related to stronger advection and vertical mixing. The vertical distribution of $NO_2$ is evaluated in section 3.3.2. Regarding advection, the wind speeds at 10 m altitude in the CM02 domain have been compared with measurements at 95 stations located within the CM02 domain (see section 3.1). A small positive bias of ca. $0.5 \, \mathrm{m\,s^{-1}}$ was found. A cold bias of $\sim 3 \, ^\circ C$ (a general feature of COSMO in summer over western Europe (Böhm et al., 2006)) was observed across the CM02 domain, but that should not cause a lower bias in the

simulated $NO_2$ VMRs.





An evaluation of surface $O_3$ VMRs with respect to the in situ measurements in a similar way as that for $NO_2$ is discussed in Appendix B.

### 3.3 Comparison of tropospheric columns

#### 3.3.1 Vertical column densities

The general approach involving an evaluation of model VCDs involves summing up simulated $NO_2$ partial columns (concentration times height of individual model grid boxes) vertically. The same approach can be applied both for evaluation of VCDs with respect to satellite and MAX-DOAS observations. However, different inferences can be drawn from these comparisons owing to the difference in sensitivity volumes. When compared to satellite observations, a larger weight is assigned to $NO_2$ at higher altitudes where satellite sensitivity is higher. Mertens et al. (2016) evaluated the $NO_2$ VCDs from the MECO(2) system

using SCIAMACHY observations and found that the model performed well in reproducing the spatial variability. In contrast to the satellite observations, MAX-DOAS measurements have higher sensitivity within the boundary layer. From the CM02 domain of all three set ups, we learn that the partial column in the lowest 1 km accounts for $\sim 80\%$ of the tropospheric $NO_2$ column.

  However, a generalized vertical integration on a regular model grid can introduce artefacts, because MAX-DOAS measure-

ments are rather sensitive to air mass in the viewing direction for distances of up to a few kilometres. The artefacts increase with increasing spatial heterogeneity. In order to consider $NO_2$ only in the viewing direction of the four telescopes, we first linearly interpolated the simulated concentrations along the respective viewing directions of the telescopes (see, e.g., Figure 1). The tropospheric VCDs are calculated in the following two steps: (1) summing up the partial VCDs vertically up to a height of 4 km and (2) taking the mean of VCDs up to a fixed distance (3 km as a first estimate) only in the line of sight of the

telescope. An example time series of the simulated VCDs for the $UBA_{di}$ set up is shown in Figure 6 along with the measured MAX-DOAS VCDs.

  Figure 7 shows the agreement between the MAX-DOAS geometric VCDs and the simulated VCDs as scatter plots for the three different set ups in different panels. The frequency distribution of the measured and the simulated VCDs are also shown next to the corresponding panels.

We observe a large scatter between the MAX-DOAS and simulated VCDs in all three model set ups, but the best agreement was observed for the $UBA_{di}$ set up with 78% of the simulated VCDs no less than half and no greater than twice the magnitude of the measured VCDs and the best Pearson correlation coefficients ($R = 0.33$) among the three. For $TNO_{fl}$, large underestimation of VCDs was observed, as also seen for surface VMRs in section 3.2. The markedly lower bias (see Table C2 and C3) can be attributed to significantly lower input $NO_x$ emissions (e.g., $\sim 40\%$ lower compared to UBA emissions over Germany for 2018).

Using UBA emissions reduces the bias from 37 – 47 % to 9 – 21 % for the different telescopes. Adding diurnal variability to emissions reduces the bias further (-1 – 13 %), as it increases the emissions during the daytime (see Figure A1 and Table C2). While the model was able to capture the general trend in day-to-day variability, the intra-day variability could not be reproduced on most of the days. The agreement was much better on the days with clear sky conditions (4–9 May) and periods





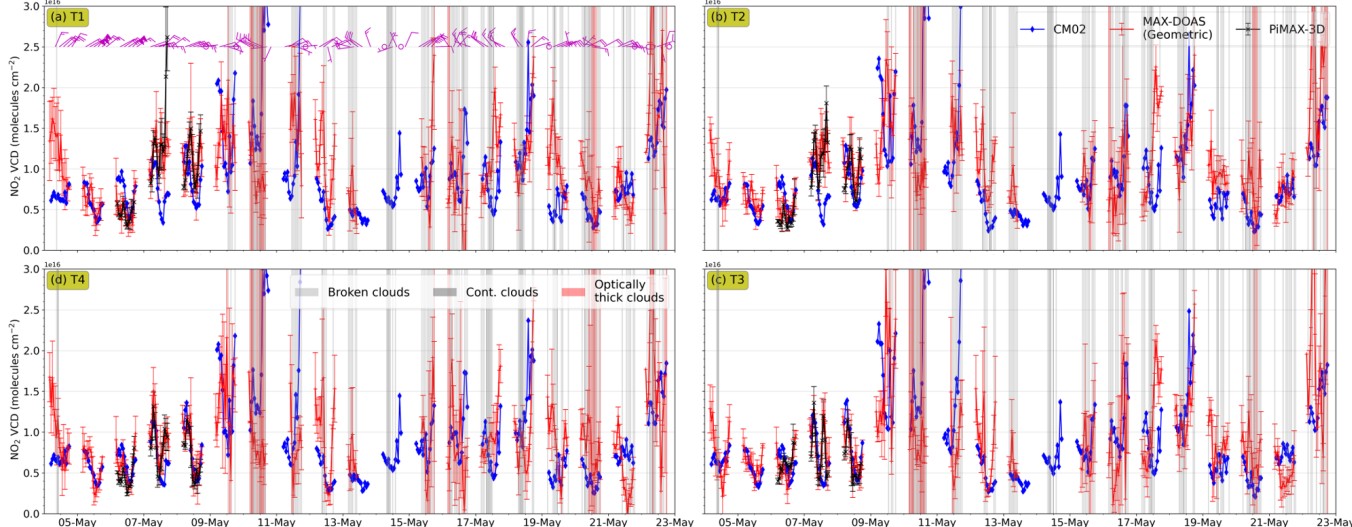

**Figure 6.** Comparison of $NO_2$ VCDs along the viewing directions of the four telescopes (T1-T4) simulated using the $UBA_{di}$ set up in the CM02 domain. MAX-DOAS VCDs are retrieved using geometric approximation (red markers) for the complete measurement period and using $\pi$-MAX for three days (black markers). The error bars represent the hourly $1\sigma$ standard deviations. The shaded regions in the time series represent various categories of clouds as described in the legend. The purple barbs in the panel A) show the measured wind speed and wind directions, with half arrows, full arrows and flags corresponding to wind speeds of 1, 2 and 5 m s$^{-1}$, respectively.

of other days with cloud-free conditions. From Tables C2 and C3 we note that the $R$ and RMSD values improve from 0.27–0.39 and 57–67 %, respectively in the $UBA_{di}$ set up for all measurements of all four telescopes to 0.37–0.52 and 50–53% if the analysis is restricted to cloud-free conditions. Between 5 and 9 May, the simulated VCDs matched almost exactly to the MAX-DOAS VCDs for the telescopes T3 and T4, but for T1 and T2 the agreement was not as good. Several factors can contribute to the observed differences, but there are at least two shortcomings related to VCD comparison, which hinder a conclusive assessment.

– The MAX-DOAS VCDs are calculated using the geometric approximation which assumes a single scattering event of the incoming photons above the trace gas layer. This yields reasonable VCDs for clear sky conditions with a low aerosol load scenario (Shaiganfar et al., 2011; Kumar et al., 2020). More accurate VCDs can be retrieved using the profile inversion approach, which also accounts for aerosol extinction profiles and the relative sun geometry. For the three clear sky days with low aerosol load, we also performed trace gas profile inversion using the sophisticated $\pi$-MAX approach, which also considers a linear change in $NO_2$ concentration along the line of sight. The VCDs retrieved using $\pi$-MAX (shown in Figure C1) agree quite well with the geometric VCDs for these days. For most of the profile inversion algorithms used currently (Frieß et al. (2019) and references therein), it is assumed that trace gases are homogeneously distributed and that MAX-DOAS is equally sensitive within the horizontal sensitivity distance; this can be an additional source of error. Previous studies (e.g. Blechschmidt et al. (2020) and Vlemmix et al. (2015)) used the optimal estimation based profile



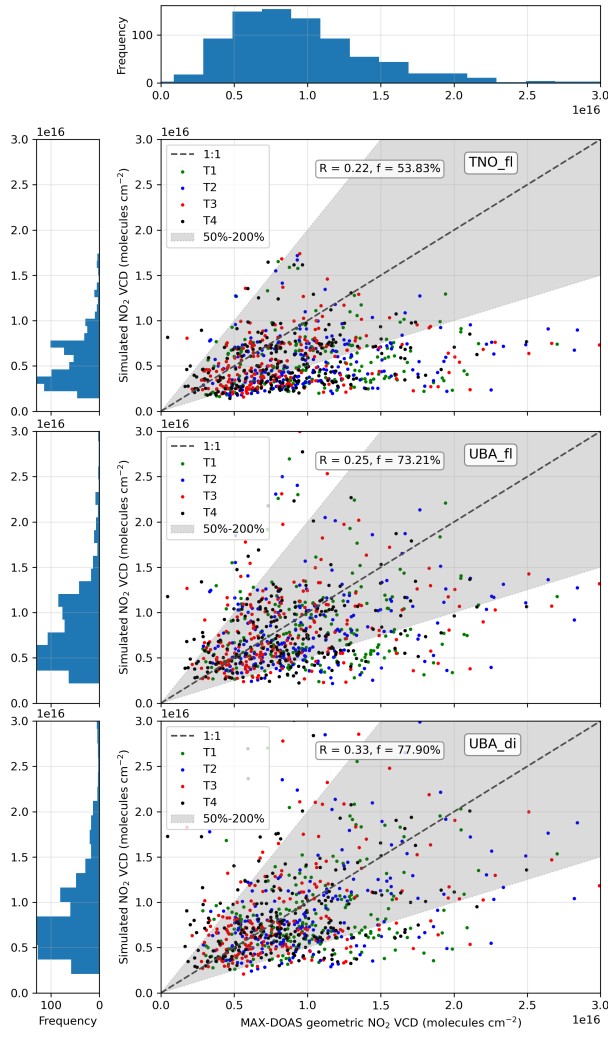

**Figure 7.** Scatter plot of simulated VCDs against measured VCDs for all four telescopes combined for 30° elevation angle. Individual points correspond to the hourly mean VCD values. The top, middle and bottom panels correspond to $TNO_{fl}$, $UBA_{fl}$ and $UBA_{di}$, respectively. $R$ represents the Pearson correlation coefficient, and $f$ represents the fraction of simulated $NO_2$ VCDs no less than half and no greater than twice the magnitude of geometric VCDs. The frequency distribution of the measured VCDs are shown above the first panel and those for the simulated VCDs from the three set ups are shown on the left of the respective scatter plots.

inversion approach, which also requires an a priori estimate of the $NO_2$ vertical profile and can bias the model evaluation
       if the assumed a priori profile is similar to that simulated by the model. The averaging kernels ($A_k$) can be applied on
       the model partial column ($V_k$) to calculate the modified VCD ($VCD_{corr}$), using the a priori profile $V_k^a$ according to





equation 3, which can be directly compared to the MAX-DOAS VCDs.

$$VCD_{corr} = \sum_k \widehat{V_k} = \sum_k V_k^a + A_k(V_k - V_k^a) \tag{3}$$

For high $A_k$ (i.e., close to 1 ), $\widehat{V_k}$ is limited by the simulated $V_k$, whereas for low $A_k$ (i.e., close to 0; where MAX-DOAS sensitivity is limited), $\widehat{V_k}$ is limited by the a priori profile. Hence, the choice of the a priori profiles can impact the comparison in the latter scenario.

– While calculating the simulated VCDs, model $NO_2$ fields are given equal weights up to a distance of 3 km in the viewing direction and 4 km altitude. As mentioned previously, distances in both dimensions were only a first estimate
and the actual MAX-DOAS sensitivity distances in both dimensions might vary according to the aerosol load, trace gas distribution, viewing geometry with respect to the sun, and presence of clouds. Moreover, even within the actual sensitivity volume, the sensitivity might vary as a function of distance from the telescope. Blechschmidt et al. (2020) have previously demonstrated that accounting for vertical sensitivity of MAX-DOAS (via averaging kernels) does not noticeably affect the simulated VCDs, because most of the $NO_2$ is located within the boundary layer and the averaging
kernel profile has a similar shape as the model $NO_2$ vertical profiles. Hence, vertical sensitivity was not an issue where most of the $NO_2$ is located. However, if the profile is elevated, then this may no longer hold true. Nevertheless, sensitivity in the horizontal direction still needs to be accounted for, as large heterogeneity is expected close to the emission sources for short-lived species like $NO_2$. The studies by Blechschmidt et al. (2020) and Vlemmix et al. (2015) proposed the relatively coarser model resolution of up to $7 \times 7$ km$^2$ as one possible reason for this discrepancy. For comparison with
ground-based measurements (e.g., MAX-DOAS), it is crucial to have model simulations with a grid resolution finer than or the same as the typical sensitivity ranges of the measurements. If that is not the case, spatial heterogeneity within the model grid box would result in underestimation of the enhancement and overestimation of the background values due to spatial smoothing. For MAX-DOAS measurements at 30° elevation angle, the horizontal sensitivity distance (HSD) can be approximated using the boundary layer height (BLH) ($HSD = BLH/sin\alpha$), which would be in the range of 1-3 km
in the daytime. However, the exact HSD also depends on the aerosol conditions, which can vary significantly over time and should be retrieved from measurements.

In the next section, we will address these shortcomings by calculating the differential slant column densities ($dSCD$s) using the simulated $NO_2$. Simulated $dSCD$s can be directly compared to the corresponding quantities derived from the DOAS analyses while also avoiding several assumptions and approximations discussed earlier. The $dbAMF$s used for $dSCD$ calculation
inherently account for the aerosol conditions and hence also address issues related to spatial sensitivity (see section 2.2.2). This also provides a way for quantification of the horizontal sensitivity distances of MAX-DOAS measurements.



### 3.3.2 Slant column densities

**Calculation of simulated dSCDs**

For calculation of $dSCD$s from the model simulated $NO_2$ fields, we mimic the viewing geometry and sensitivity volume
corresponding to MAX-DOAS measurements using the differential box air mass factors as described in section 2.2.2. Using
MAX-DOAS, we probe the vertical and horizontal variation of $NO_2$ concentrations by measuring at various elevation angles.
The sensitivity of the MAX-DOAS measurements at a given elevation angle (EA) is described by the differential box air mass
factors ($dbAMFs$). An example of the $dbAMFs$ along the viewing direction of telescope T1 for `09-05-2018 14:00`
UTC for EAs ranging from 3 to 30° is shown in Figure 8. In each viewing azimuth (corresponding to T1-T4) and for each EA
($\alpha$; between 2 and 30°) we perform a two-dimensional summation of partial VCDs ($V_{i,k}$) weighted by the differential box air
mass factors ($dbAMF_{\alpha,i,k}$)(unit-less), along the distance from MAX-DOAS (indexed as $i$) and altitude above the instrument
(indexed as $k$):

$$dSCD_\alpha = \sum_{i,k} V_{i,k} \times dbAMF_{\alpha,i,k} \qquad (4)$$

with

$$V_{i,k} = c_{i,k} \times dh_{i,k} \qquad (5)$$

Where, $c_{i,k}$ and $dh_{i,k}$ represent the concentration ($\mathrm{molecules\,cm^{-3}}$) of trace gas in the grid with a thickness of $dh_{i,k}$ (cm).

Using equation 4, we can estimate the EA dependent horizontal sensitivity distances (HSD) in the viewing direction of the
MAX-DOAS as the distance from the instrument which accounts for 90% of the simulated $dSCD$s. Fig C3 shows the HSD
for all the off-axis elevation angles for the four telescopes. The mean HSD increases from 3-4 Km for 30° EA to 8-9 Km for
3° EA. In contrast to the comparison of the geometric VCDs in the previous section, which is limited to only one EA, we can
evaluate the $dSCD$s at various EAs with varying sensitivity volume from the location of the instrument. Additionally, while
calculating the $dSCD$s, we also account for the horizontal heterogeneity and varying sensitivity within the sensitivity volume
in the viewing direction.

Figure 9 shows an example time series of measured and simulated $dSCD$s for $UBA_{di}$ for 30° EA. Comparing these values
to the VCDs shown in Figure 6, we observe that the simulated $dSCD$s are higher than the calculated VCDs as shown in the
previous section, implying $dAMF$s at 30° EA are larger than 1. This is also observed under the cloud-free conditions, which
corroborates the drawback in the geometric approximation and assumption of spatial homogeneity in the viewing direction.

**Evaluation of simulated dSCDs**

Figure 10 shows the distribution of measured and simulated $dSCD$s (different colours for the different model set ups described
in Table 1) for the various elevation angles as box and whiskers plots for the four telescopes in separate panels. Measurements
performed at low EAs have large light paths and are more sensitive to air mass close to the surface (higher $dbAMF$s; see





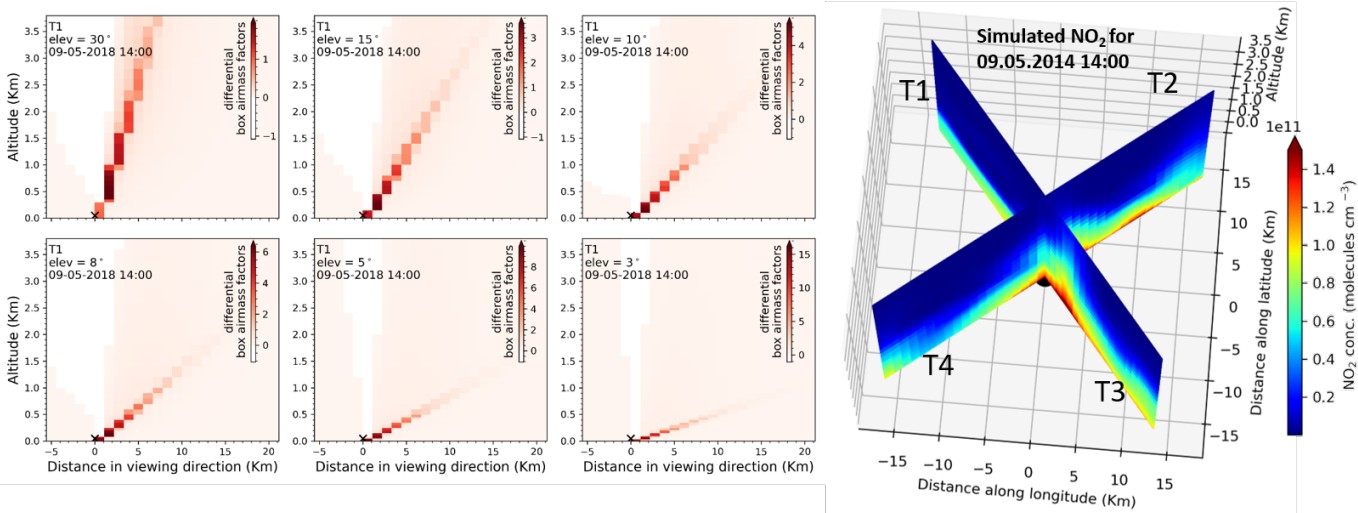

**Figure 8.** Left) The differential box air mass factors ($bAMFs$) for 30, 15, 10, 8, 5 and 3° elevation angles retrieved using 3D profile inversion of measured $O_4$ $dSCDs$ along the direction of telescope T1 . Please note the different colour scales of differential $dbAMFs$ used for the different elevation angles. Right) Simulated $NO_2$ concentrations from the $UBA_{di}$ set up in the CM02 domain from surface up to an altitude of 2.5 km along the viewing directions of the four telescopes T1, T2, T3 and T4 on 09-05-2018 14:00 UTC.

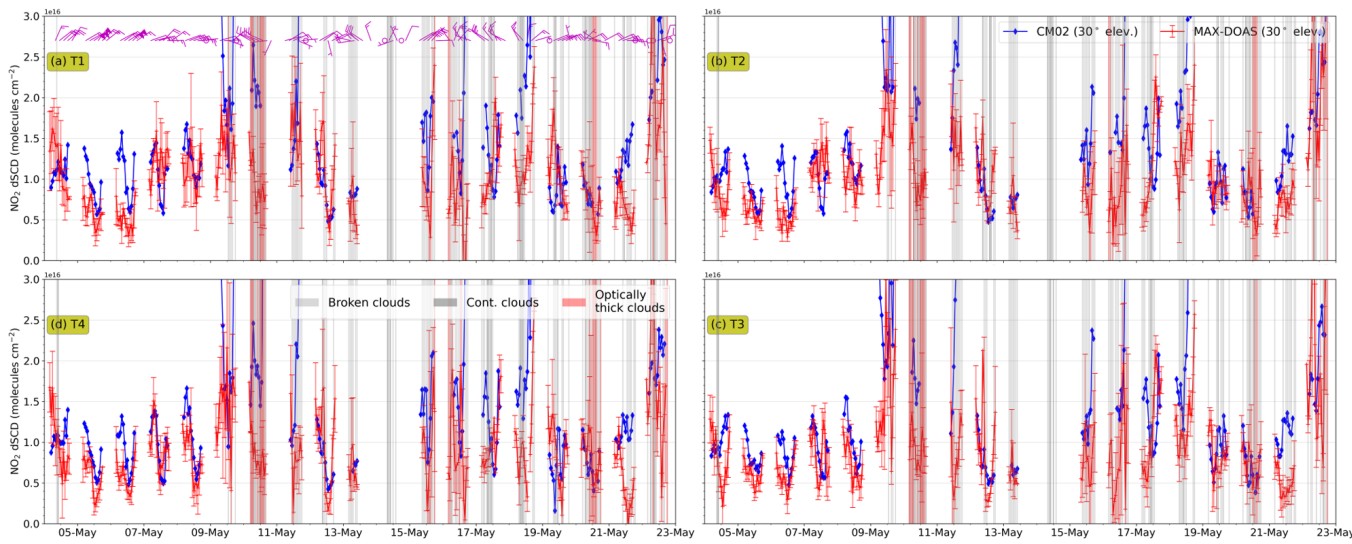

**Figure 9.** Comparison of $NO_2$ differential slant column densities at 30 ° elevation angle of the four telescopes simulated using $UBA_{di}$ in the CM02 domain. The shaded regions in the time series represent various categories of sky conditions as described in the legend.

Figure 8), and hence larger $dSCDs$ are observed for the low EAs. Surprisingly, we did not observe major differences in the measured as well as simulated $dSCDs$ among the four telescopes, besides slightly higher values for T2, which points towards





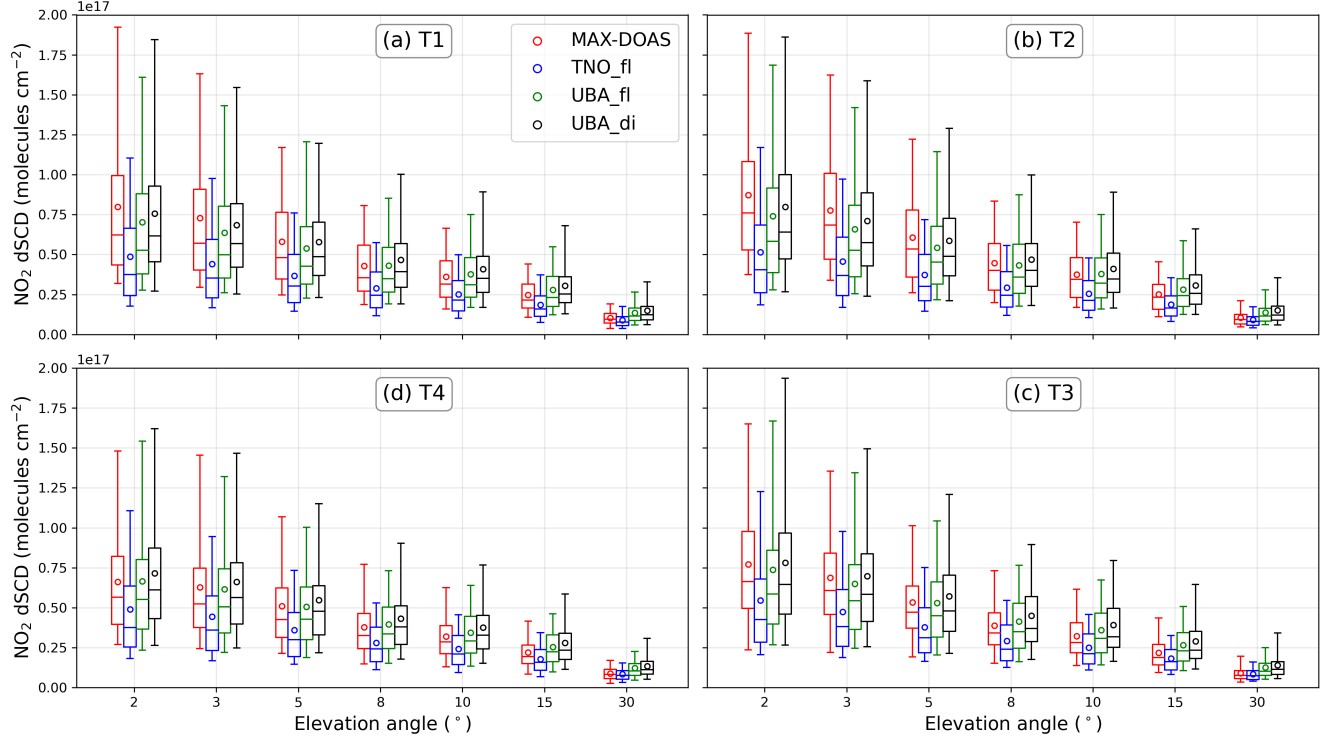

**Figure 10.** Box and whiskers plots for the different telescopes (panel a) - d) ) showing the distribution of measured and simulated $dSCD$s for the three different model set ups (depicted in different colours) for various elevation angles ($2°, 3°, 5°, 8°, 10°, 15°$ and $30°$).

the city centre of Mainz and lower values for T4, which spans mostly agricultural lands for a distance of 10 km. This can

partially be explained by the prevailing wind directions (panel (a) of Figure 6), as for most of the time, easterly winds bring the air mass from the urban locations to the agricultural lands.

   Figures 9 and 11 show example time series of measured and simulated $dSCD$s (for UBA$_{di}$) for $30°$ and $3°$ EAs, respectively. Various statistical parameters corresponding to the agreement between MAX-DOAS and model simulation for individual telescopes and EAs are summarized in Table C2 for all measurements and in Table C3 for cloud-free cases only. For $30°$ EA we did

not observe any significant change in the agreement of temporal variability (i.e. $R$) between model and MAX-DOAS, whether we compare $dSCD$s or VCDs. However, significantly more information is gleaned with respect to the available $dSCD$s corresponding to all the off-axis EAs at which measurements were performed. Instead of comparing only one VCD value for a complete elevation sequence, we include dSCDs corresponding to each of the elevation angles. For example, in Figure 11, we observe a much better agreement between the measured and simulated $dSCD$s at $3°$ EA, especially on the cloud-free days.

In Figure 12, we show the comparison of measured and simulated $dSCD$s for the three different model set ups in different panels for all the EAs and all the telescopes. The frequency distribution of the measured $dSCD$s at various EAs are shown above the top panel, while those for the simulated $dSCD$s are shown in the panels left of the scatter plot of the corresponding





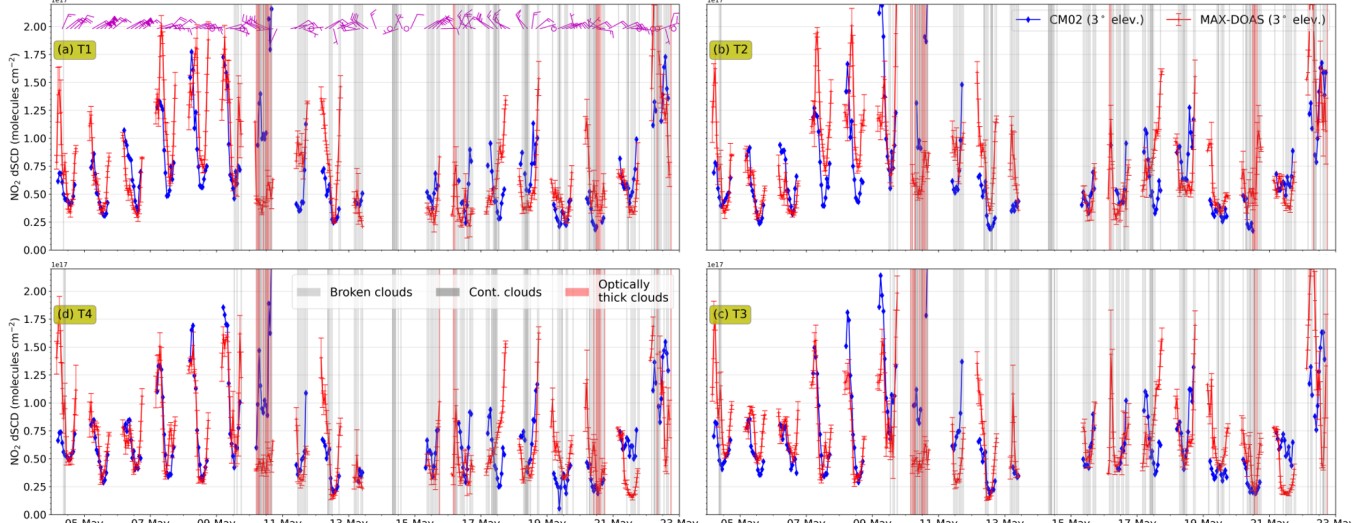

**Figure 11.** Same as Figure 9 but for $3°$ elevation angle

model set ups. We observe a good correlation between the measured and simulated $dSCD$s ($R$ = 0.63, 0.62 and 0.67 for $TNO_{fl}$, $UBA_{fl}$, and $UBA_{di}$, respectively), which further improves ($R$ = 0.66, 0.65 and 0.74) if the comparison is restricted

to cloud-free cases only. Similar to the VMR and VCD comparisons, the best accountability was observed for $UBA_{di}$, with $\sim$ 82% for the simulated $dSCD$s no less than half and no greater than twice the magnitude of the measurements for cloud-free cases. The frequency distribution of the measured $dSCD$s follows a right-skewed normal distribution for all the EAs, which is also represented best by the $UBA_{di}$ set up. The width of the peak of the frequency distribution broadens from high to low EAs, representing a larger scatter in the measurements at low EAs. For $UBA_{fl}$, the extreme values (less than half or more than

double of measurements) were mostly observed for cloudy sky conditions. Since clouds are not considered in the radiative transfer simulations and therefore also not in $dAMF$ retrieval, these only affect the measured $dSCD$s. The improved agreement between simulated and measured $dSCD$s for the cloud-free conditions are more obvious for the individual EAs (see Tables C2 and C3) with significantly higher Pearson correlation coefficients and smaller RMSD values.

From Tables C2 and C3, it can be inferred that the agreement between measured and simulated $dSCD$s improves at lower

EAs, which also indicates a better performance of the model in the layers close to the surface. For example, both for the $UBA_{fl}$ and $UBA_{di}$ set ups, a large positive bias in the range $27-42\%$ was observed at $30°$ EA, but for low EAs (e.g. $\leq 8°$) small biases in the range of +7 and -15% were observed for all the azimuth directions in the cloud-free cases. However, more pronounced negative biases were observed for $TNO_{fl}$ simulation even at low elevation angles. A positive bias at $30°$ EA is most likely due to the stronger vertical mixing in COSMO/MESSY as also indicated by Mertens et al. (2016). Due to the stronger mixing, a

large fraction of $NO_2$ reaches at higher altitudes in the model, which has a strong weight for the $dSCD$ calculation at high EAs.





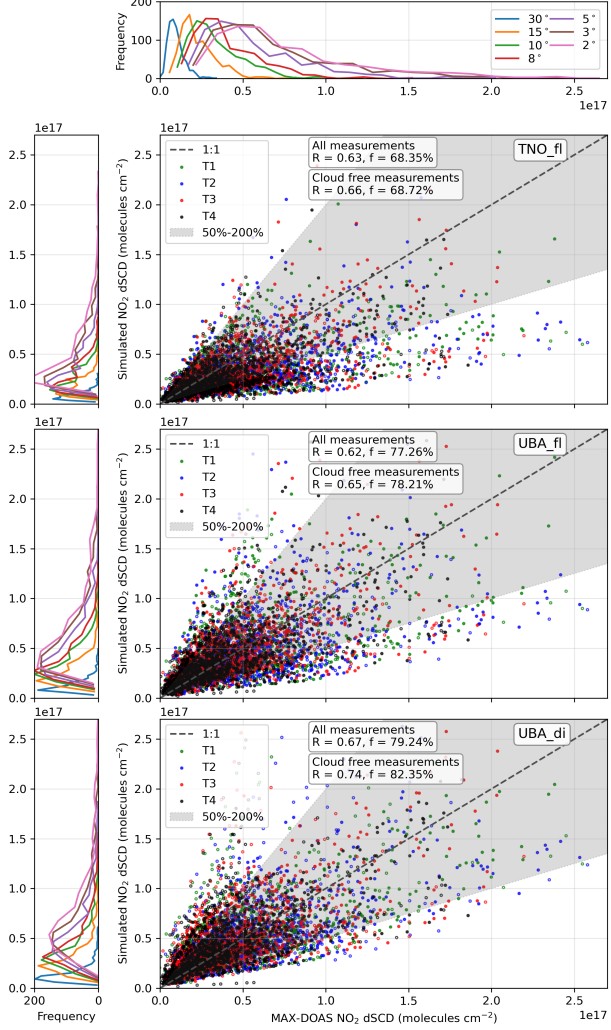

**Figure 12.** Scatter plot of simulated $dSCD$s against measured $dSCD$s for all the four telescopes combined for the elevation angles 2, 3, 5, 8, 10, 15 and $30°$. The filled circles correspond to the cloud-free scenarios, while the empty circles correspond to the cloudy conditions. The Pearson correlation coefficient ($R$) and the fraction of simulated dSCDs within 50-200% of the measurements ($f$) are annotated in the respective panels in grey for all measurements and in black for cloud-free conditions only. The top, middle and bottom panels correspond to $TNO_{fl}$, $UBA_{fl}$ and $UBA_{di}$ respectively. The frequency distribution of the measured $dSCD$s is shown above the first panel, and those for the simulated $dSCD$s from three set ups are shown on the left of the respective scatter plots.

Previous studies in which models were compared with MAX-DOAS indicated that the major disagreements arise due to weekday and weekend differences and inappropriate representation of the diurnal cycle of emissions (Blechschmidt et al., 2020; Vlemmix et al., 2015). We use the measurements at a low elevation angle (e.g. $3°$) to investigate if the model can

reproduce the measured diurnal profiles and weekday-weekend differences of $NO_2$ $dSCD$s. A low elevation angle was chosen



because the corresponding measurements have higher sensitivity close to the ground where a major fraction of $NO_x$ is emitted from sectors which show strong diurnal variability.

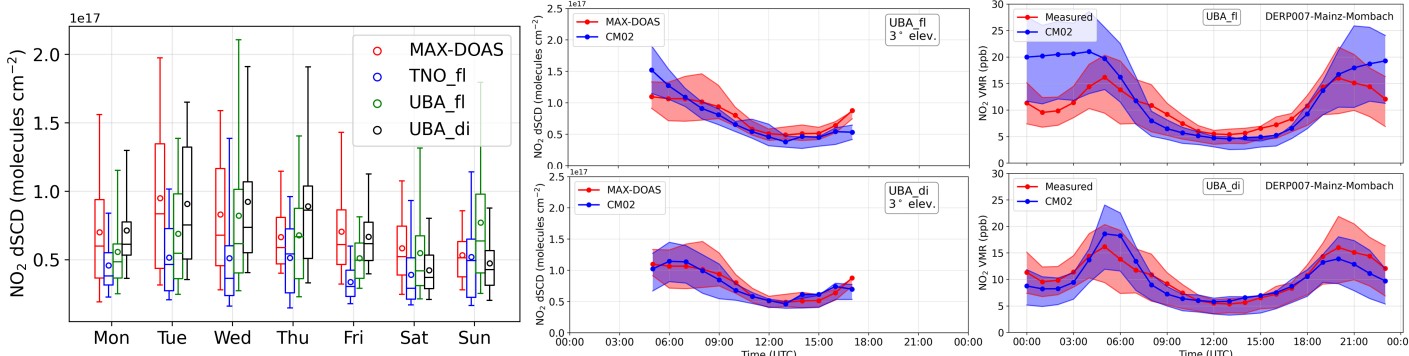

**Figure 13.** Left) Box and whiskers plot showing the distribution of $NO_2$ $dSCD$s at $3°$ elevation angles for MAX-DOAS and the three different model set ups for different days of the week. Centre) Mean hourly diurnal profiles of measured and simulated $NO_2$ $dSCD$s at $3°$ elevation angle for T1 for $UBA_{fl}$ (top) and $UBA_{di}$ (bottom) set ups. The shaded region above and below the mean line denote the $75^{th}$ and $25^{th}$ percentiles, respectively. Right) Diurnal profiles of measured and simulated $NO_2$ surface VMRs measured at a background site Mombach ($\sim 3.5$ km north of MPIC) for $UBA_{fl}$ (top) and $UBA_{di}$ (bottom) simulations.

The left panel in Figure 13 shows the distribution of measured and simulated $NO_2$ $dSCD$s at $3°$ EA, for all telescopes combined, binned according to the day of the week. The diurnal profiles of the measured and simulated $dSCD$s are shown
in the middle panel of Figure 13 for $UBA_{fl}$ in the top panel and for $UBA_{di}$ in the bottom panel. A similar plot for $NO_2$ surface VMR at the background site Mombach ($\sim 3.5$ km north of MPIC) is shown in the right panel. The lowest measured $dSCD$s are observed on the weekends. However, such a distinct weekday-weekend difference was not observed for $TNO_{fl}$ and $UBA_{fl}$ simulations, supporting the fact that the observed differences were primarily because of change in emissions and not due to varying meteorological conditions. Smaller weekend $dSCD$s were only for the $UBA_{di}$ set up which also accounts
for weekday and weekend differences in $NO_x$ emission from the transport and residential combustion sectors. Concerning the diurnal variation, both $UBA_{fl}$ and $UBA_{di}$ show similar $dSCD$s and surface VMRs during the daytime which also agrees well with the measurements. However, stronger discrepancies are observed during early morning and late evening hours. Comparison of surface VMRs provide further information about the night-time, where even a larger difference is observed between the two model set ups. $NO_2$ has a short lifetime of a few hours in the daytime, which together with a strong mixing in
the daytime boundary layer compensates even for $\sim 50\%$ higher emissions (an increase in emissions from the transport sector by a factor of two would translate to an overall increase of $\sim 50\%$ in $NO_x$ emissions). The use of diurnal emission profiles has the strongest effect on the simulated VMRs and $dSCD$s when the atmospheric lifetime of $NO_2$ is long. From Figure 13, it is evident that the use of diurnally varying emissions is necessary to reproduce the observed diurnal variability of $NO_2$. For the complete time series of $dSCD$s, we also observe much improved Pearson correlation coefficients in the range of 0.5 and
0.8 if diurnal and day-of-the-week variability in emissions are considered in UBA emissions as compared to those simulated





using "flat" emission profiles (see Table C3). For model studies concerning satellite measurement having afternoon overpasses (e.g. OMI, TROPOMI), consideration of these factors for the anthropogenic emissions will have a relatively weak effect in our study domain or similar urban environments. For the biomass burning regions (e.g. tropical forests, southern Africa), strong $NO_x$ emissions are typically observed during mid-day, which results in a diurnal profile of $NO_2$ columns with a broad daytime peak (Boersma et al., 2008). Consideration of diurnal profiles of emissions in model simulations is crucial for comparisons with satellite observations in these regions (Miyazaki et al., 2012; Boersma et al., 2008).

## 4   Conclusions

We performed high spatial resolution (up to $2.2 \times 2.2$ km$^2$) regional model simulations focused on south-west Germany to evaluate the short-lived pollutant $NO_2$ using MAX-DOAS and a network of in situ measurements. Three different MECO(3) simulations were performed in order to investigate the importance of model spatial resolution and the influence of spatial and temporal resolution of input emission inventories. Anthropogenic emission inventories generally used for model simulations (e.g. TNO MACC for Europe) do not cover the most recent periods in most of the cases and are available at spatial resolutions coarser than that of the model. We show that the spatial patterns of $NO_2$ are best reproduced in the UBA$_{fl}$ set up, in which an up to date and high-resolution ($1 \times 1$ km$^2$) input emission inventory (UBA) is used. In the UBA$_{di}$ set up, use of accurate temporal profiles (e.g. diurnal and day of the week) of the road transport and residential combustion emissions improves the agreement of the temporal profiles at individual measurement stations. An improved agreement was observed at the background and industrial locations with an overall bias of less than 10% for the UBA$_{di}$ set up. However, the model largely underestimates the $NO_2$ VMRs (by up to 50%) at the traffic-adjacent locations. For the background locations, the mean diurnal profiles were accurately simulated in the UBA$_{di}$ set up. Biases were stronger if the fine resolution emissions were used for a coarser resolution model simulation (e.g., $7 \times 7$ km$^2$). In contrast, a finer resolution model employing a coarse emission inventory did not result in the addition of large spatial details.

We employed the measurements of a 4-azimuth MAX-DOAS instrument from Mainz to first compare the tropospheric $NO_2$ VCDs. The day-to-day variability was reasonably well reproduced by the model in the UBA$_{di}$ set up for all four viewing directions with biases of between -10% and 2%. To further augment the surface VMR and tropospheric VCD evaluation, we apply a consistent approach of comparison of the so-called differential slant column densities ($dSCDs$), which, we suggest, comes with several advantages. Firstly, $dSCD$s are available for a number of elevation angles, each of which has distinct sensitivities to the different vertical level of the troposphere. Hence, this approach enables an evaluation of the vertical distribution of $NO_2$ in the model. Additionally, the horizontal sensitivity distance of the MAX-DOAS instrument also changes for different elevation angles, as described by the corresponding differential box air mass factors. Hence, when using the measurement at one single location, model evaluation can be performed for various sensitivity volumes. Secondly, as compared to the VCD comparison, which gives one comparable quantity per complete elevation sequence of MAX-DOAS measurements, dSCDs provide a way to evaluate the simulation against the measured values for each elevation angle. Finally, for the $dSCD$ compar-





ison, we succeed in overcoming the uncertainties introduced by the assumption of a homogeneous spatial distribution of $NO_2$ and in some cases a priori estimates of its vertical distribution for the retrieval of VCDs from the MAX-DOAS measurements.

We evaluated the simulated dSCDs in the four azimuth direction for seven elevation angles (EA) ranging from $2°$ to $30°$. We observe a similar variation of measured and simulated dSCDs for various EAs indicating a reasonable vertical distribution of $NO_2$. The agreement between model and simulation improved for lower elevation angles indicating better accountability at near-surface layers. The agreement further improved, if only the measurements in cloud-free conditions were considered for the comparison. We did not observe large differences in the measured dSCDs in the four azimuth direction because of

the prevailing wind direction from urban areas. We also show that the consideration of diurnal profiles of the anthropogenic emissions is crucial for comparison with $NO_2$ dSCDs and VMR measurements for early morning and night time hours. For the afternoon hours, however, even up to 50% higher anthropogenic $NO_x$ emissions only have a minor effect on ambient VMRs and dSCDs due to its short atmospheric lifetime.

Over the last two decades, several MAX-DOAS measurements have been reported from locations across the world, which

have substantially contributed to the evaluation of satellite observations. We think that the complexity and uncertainties involved in VCD retrieval from the MAX-DOAS measurements have so far hindered a similar scale application in model evaluation. The consistent $dSCD$ comparison approach proposed in our study validates such usage of these valuable datasets, which can be used to evaluate the vertical distribution of trace gases within the boundary layer.

*Code availability.*    The Modular Earth Submodel System (MESSy) is a multi-institutional project, and its license is available to the affiliates

of these institutions. Affiliation to MESSy can be obtained by signing the MESSy Community End-User Licence Agreement (EULA) and accepting the MESSy Software Licence Agreement (SLA) available at www.messy-interface.org. ECHAM5 model is available under the Software Licence Agreement of the Max Planck Institute for Meteorology, Hamburg. The COSMO model can be obtained either by institutional license provided by the German Weather Service (DWD) or individual license provided by the CLM-Community. Further details about the CLM community is available at https://wiki.coast.hzg.de/clmcom.

*Data availability.*    TNO MACC III emission data (Kuenen et al., 2014) is provided by Jeroen J.P. Kuenen and Hugo A.C. Denier van der Gon at the TNO, Princetonlaan 6, 3584 CB Utrecht, The Netherlands by signing the disclaimer and conditions for data use. The UBA anthropogenic emission data is made available by the German Environment Agency (Umweltbundesamt). Further details about the UBA emissions are available at www.umweltbundesamt.de/deutschland-karten-zu-luftschadstoff-daten. The hourly resolution measured data for $NO_2$ and $O_3$ is also provided by Umweltbundesamt and can be freely downloaded via the web interface www.umweltbundesamt.de/en/data/

air/air-data/stations. The meteorological data used for model evaluation are provided by the German Weather Service (DWD) via the climate data centre web interface available at https://cdc.dwd.de/portal.





## Appendix A: Temporal profiles of emissions

Both TNO MACC III and UBA emissions are available at a temporal resolution of one year. The monthly profiles of the anthropogenic emissions depend on the emitted species, emissions sectors and the country. For Europe, these factors are also

provided by Builtjes et al. (2002), and, are used to create monthly resolution emissions. Builtjes et al. (2002) and Schaap et al. (2005) also provide recommendations for sector-specific fine temporal profiles (day of the week and diurnal) of emissions (see Figure A1). The fine temporal profiles are applied only in the UBA$_{di}$ set up using the MESSy ONEMIS submodel as described in section 2.1. The recommendation of Builtjes et al. (2002) and Schaap et al. (2005), however, do not differentiate between the diurnal profiles of emission between weekdays and weekends, which is crucial for the road transport emissions.

Hence, for the road transport sector, we used the actual hourly vehicle count on the A60 motorway for 2018, and derived the temporal emission profiles assuming a direct scaling between the number of vehicles and emissions. The actual vehicle counts are provided by the automatic vehicle counter and are available at www.bast.de/BASt_2017/DE/Statistik/statistik-node.html. From Figure A1, we note that for weekdays, the derived profiles look similar to those recommended by Schaap et al. (2005). However, for the weekend, the shape of the diurnal profiles are markedly different from the factors derived using actual vehicle

counts, which show a single broad afternoon peak.

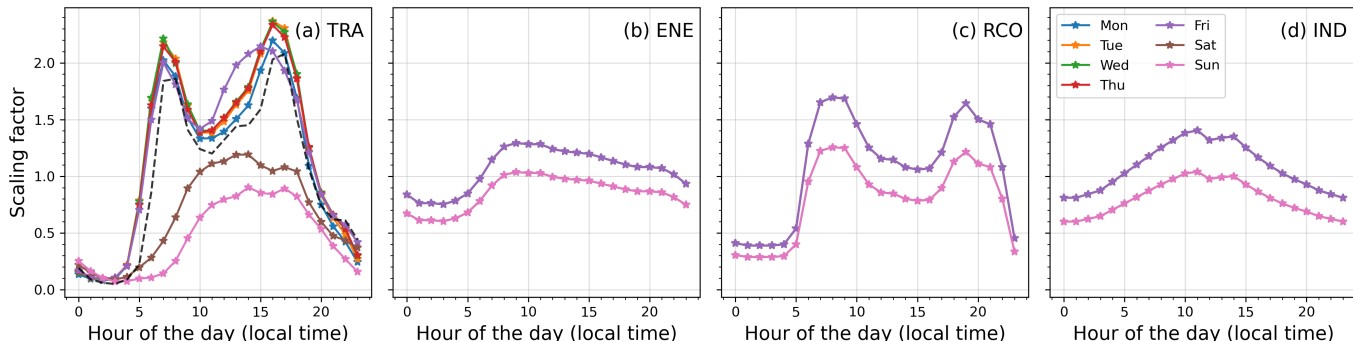

**Figure A1.** Temporal profiles of NO$_x$ emissions from various sectors. "TRA": Road transport, "ENE": Energy generation, "RCO": Residential and non-industrial combustion, "IND": Industries. For "TRA", lines and markers correspond to the profiles derived using actual vehicle count, while for others, profiles are derived according to the recommendation of Schaap et al. (2005). For "TRA", the recommendation of (Schaap et al., 2005) are shown as the dashed line in panel (a).

Please note that the current implementation of incorporation of diurnal emission factors is limited to surface emissions. Since for "ENE" and "IND" sectors, a significant fraction is emitted at high altitudes, the day of the week and diurnal profiles could not be applied to these sectors. However, from Figure A1, we note that these temporal variations are not as strong as for "TRA" ($> 200\%$ peak to peak) and "RCO" ($> 120\%$ peak to peak) sectors. Moreover, for Germany, "ENE" and "IND" account

for only ca. 14% and ca. 18% of the total NO$_x$ emissions as compared to ca. 45% from "TRA" only. Hence, we only expect a minor effect of including the day of the week and hourly temporal factors of emissions of "ENE" and "IND" sectors on the total NO$_x$ emissions.



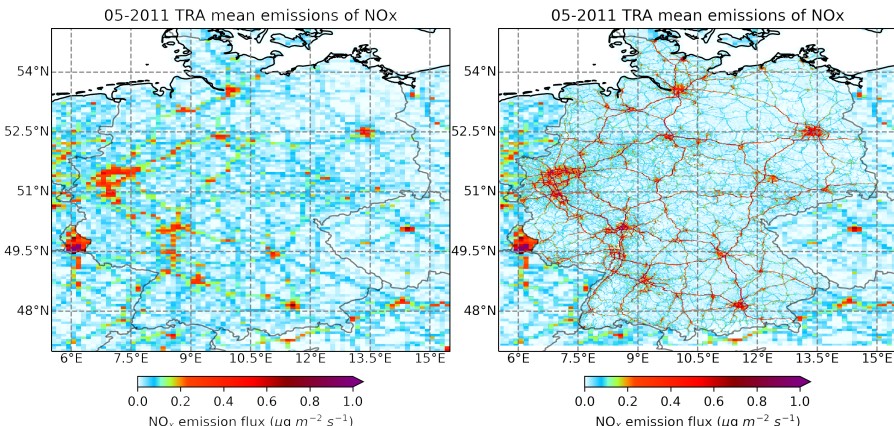

**Figure A2.** $NO_x$ emission flux from the road transport sector for left) TNO MACC III emissions for May 2011 (Kuenen et al., 2014) and right) composite UBA with Germany (May 2018) and TNO MACC III emissions outside Germany (May 2011)

## Appendix B: Evaluation of surface mixing ratios of $O_3$

Figure B1 shows the spatial distribution of the monthly mean $O_3$ VMRs and agreement with respect to the measured values

in a similar way as for $NO_2$ in Figure 4. We observe a rather smooth spatial distribution of $O_3$ as compared to that of $NO_2$, with high values relatively far from the $NO_2$ hotspots. We observe an overestimation for smaller $O_3$ VMRs for all three model set ups. Since ozone is formed photochemically in the troposphere, we have further investigated the agreement separately for daytime (07:00-18:00 UTC) and night time for the $UBA_{di}$ set up in Figure B2.

During night time, we observe a general overestimation by $\sim$ 37% for all the measurement stations combined. However,

during daytime, we observed much better agreement with a relative bias of $\sim$ -5%. Mertens et al. (2016) have previously investigated the positive bias of simulated $O_3$ for a coarser resolution ($\sim$ 12 $\times$ 12 km$^2$) MECO(n). They have shown that COSMO/MESSy simulates a relatively weaker amplitude of planetary boundary layer height (PBLH) such that the night time PBLH is biased towards higher values. This results in a large night time "reservoir" of ozone that can undergo chemical titration with $NO_x$ or dry deposited. Furthermore, a stronger vertical mixing in COSMO/MESSy brings ozone rich air, which together

with a weaker dry deposition (Travis and Jacob, 2019) causes a positive bias in simulated surface ozone in the night time. The stronger vertical mixing was also confirmed by Mertens et al. (2016) using a diagnostic tracer with no atmospheric sink. During the daytime, the cold bias of MECO(n) could also bias the rate of ozone production from precursors via rate constants. Mertens et al. (2016) have investigated this deviation by forcing the nudged EMAC simulated temperatures in the COSMO/MESSy domain. This, however, did not explain the observed bias in $O_3$.

The Taylor diagrams in Figure B3 show significantly better model performance (larger $R$ and smaller RMSD values) as compared to that for $NO_2$ in section 3.2. This is due to the relatively larger lifetime of $O_3$, owing to which strong spatial gradients are not observed. The temporal variability is, however, underestimated as evident by the relative standard deviations of less than 1. Using an updated and high resolved anthropogenic emission with high $NO_x$ did not elicit further improvement in





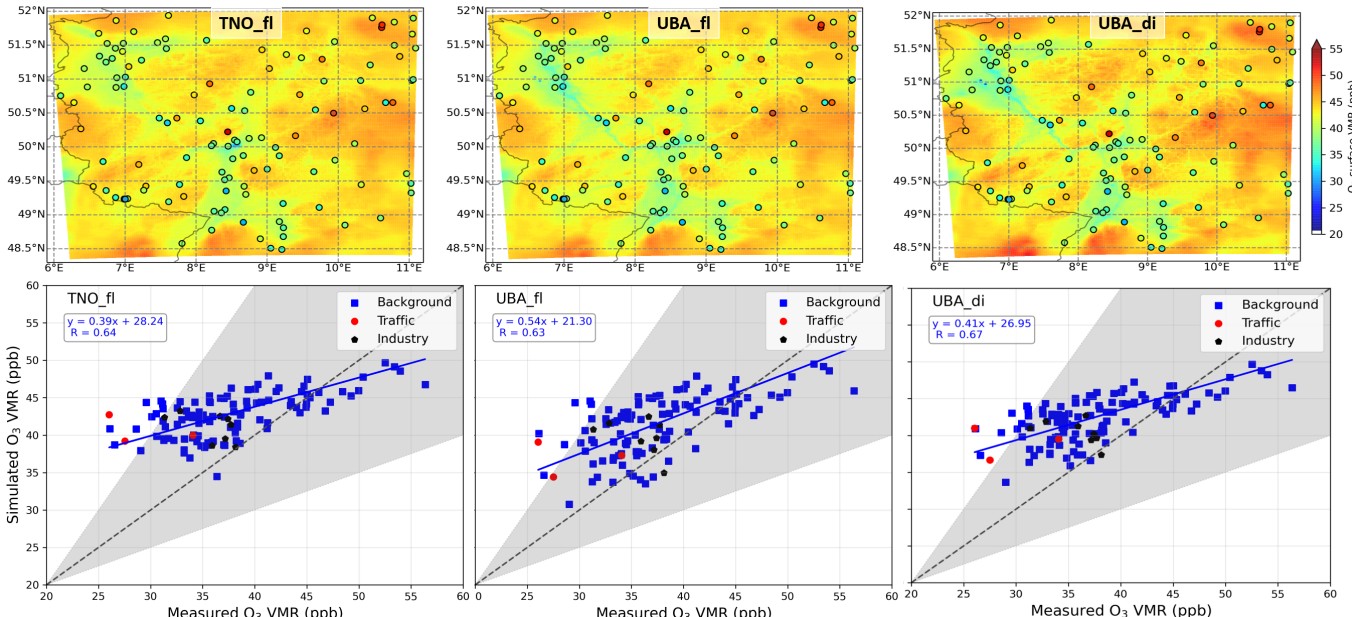

**Figure B1.** Spatial distribution of monthly mean $O_3$ surface VMRs for the three simulations using different emission inventories (left: TNO, middle: UBA without diurnal variations, right: UBA with diurnal variations) for May 2018. Bottom panel shows the scatter plot and orthogonal distance regression (ODR) weighted by the inverse of the square of the standard deviation of simulated monthly mean $O_3$ surface VMRs with respect to the in situ measured values. ODR was performed only for the background stations because of the low number of stations at the traffic-adjacent and industrial locations.

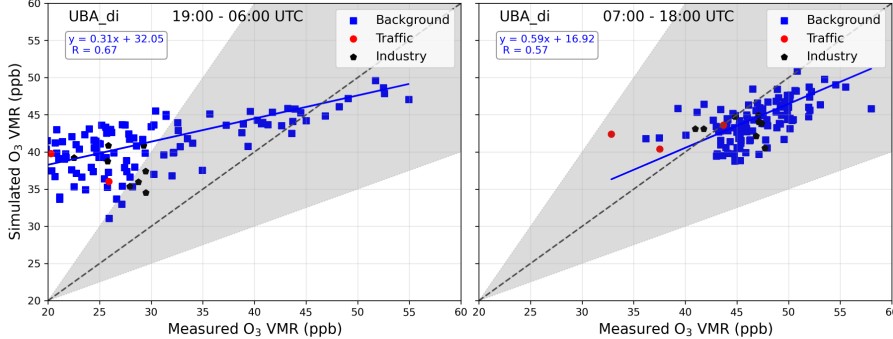

**Figure B2.** Scatter plot between measured and simulated $O_3$ for night time (left) and daytime (right) for monthly means at all the measurements stations in the CM02 domain for the UBA$_{di}$ set up.

the agreement of spatial patterns of the monthly mean measured and simulated $O_3$. The only improvement was a reduction in
overestimation corresponding to the lower values. These values are observed for the sites where generally higher $NO_2$ VMRs were observed, thus indicating a $NO_x$ saturated ozone production regime. However, there can be several discrepancies related





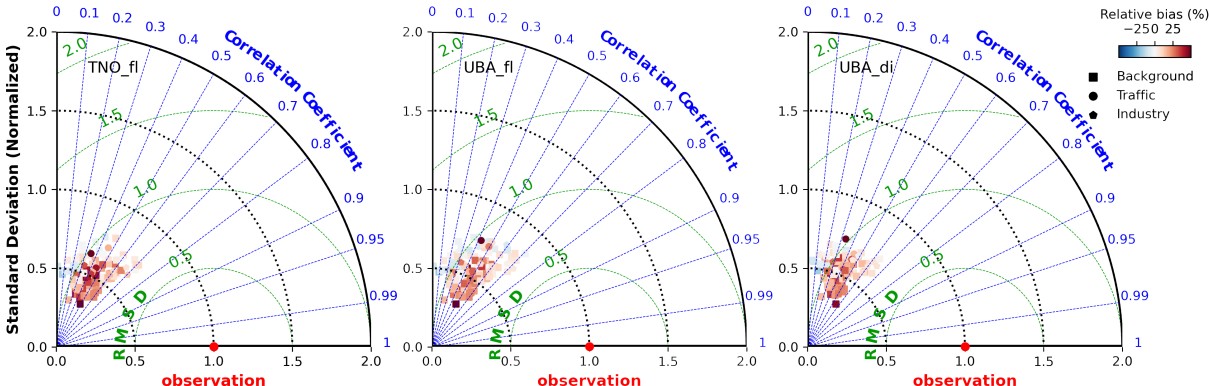

**Figure B3.** Taylor diagrams showing the Pearson correlation coefficients, normalized standard deviation and normalized root mean square difference corresponding to hourly resolved measured and simulated $O_3$ VMRs (in CM02 domain) for background, traffic, and industrial sites represented as square, circle and pentagon markers, respectively. The left, middle and right panels correspond to the $TNO_{fl}$, $UBA_{fl}$ and $UBA_{di}$ set ups, respectively.

to inappropriate NMVOC speciation, biogenic emissions and a relatively simpler chemical mechanism used in the model, due to which we are not confident about this finding. Further investigation in this direction is beyond the scope of this study and should be pursued in future works with complex chemistry.





# Appendix C: Additional figures and tables

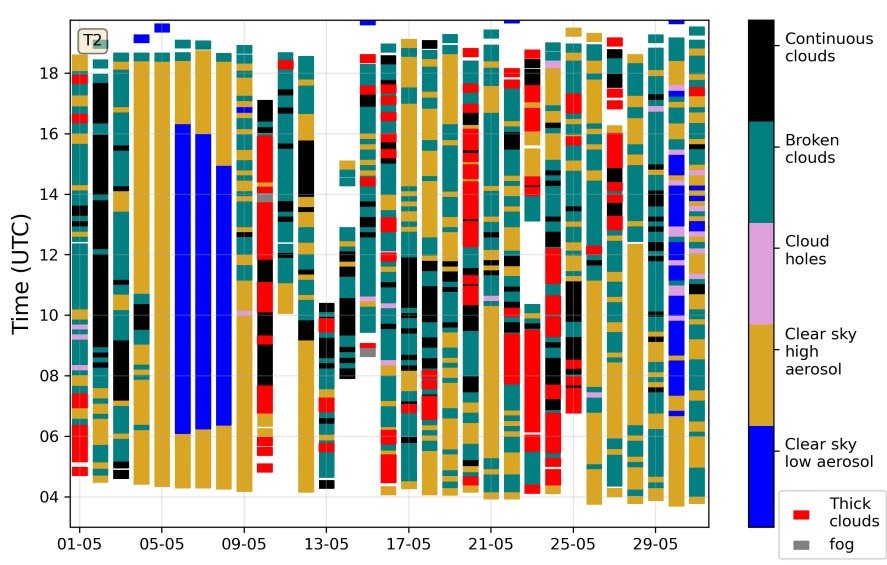

**Figure C1.** Daily variability of the sky conditions for May 2018 retrieved using MAX-DOAS measurements along the azimuth direction of
T2. Consistent sky conditions were retrieved for the other three azimuth directions

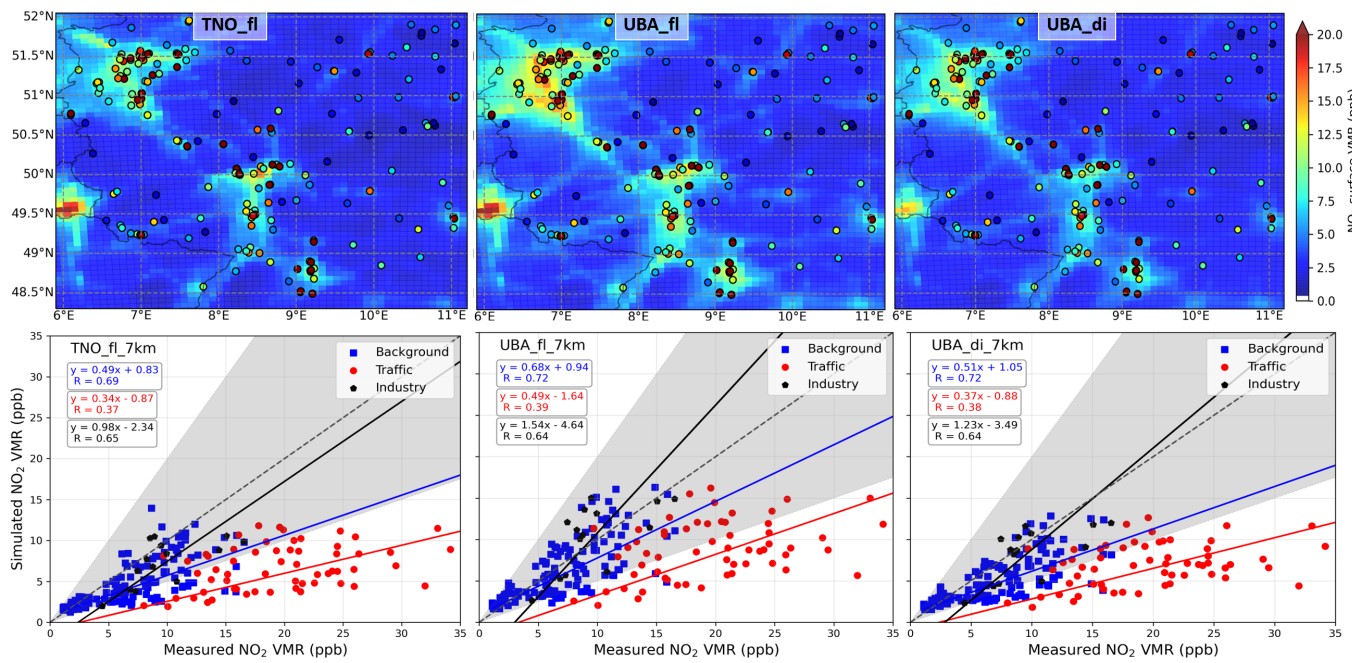

**Figure C2.** Same as Fig 4, but for the CM07 set up in the domain limited to that for CM02 boundaries.





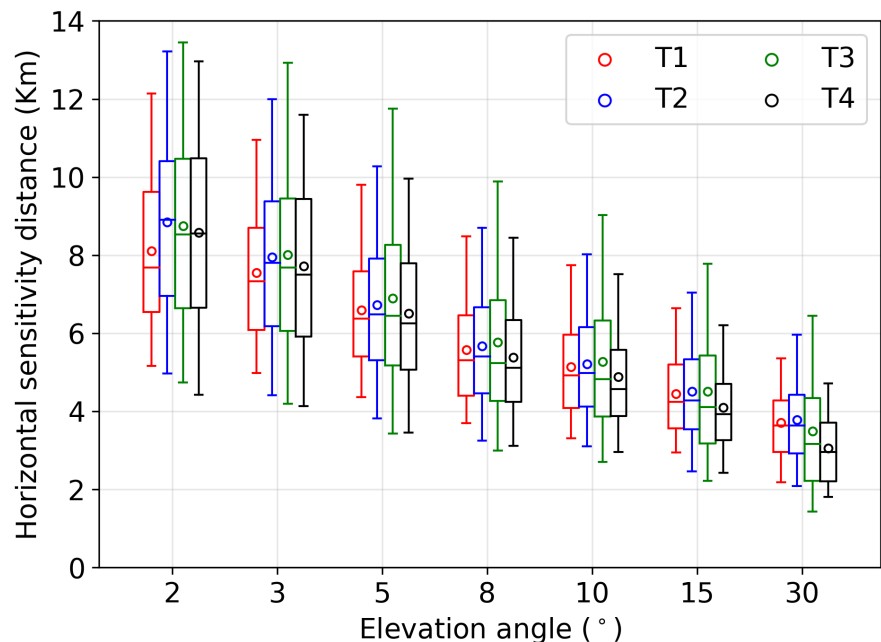

**Figure C3.** Typical horizontal sensitivity distances (HSD) for different elevation angles along the viewing direction of the four telescopes. The HSD for an elevation angle $\alpha$ is estimated as the distance along the viewing direction which accounts for 90% of simulated $dSCD_\alpha$.

**Table C1.** Fit settings chosen for MAX-DOAS spectral analyses

|  | $NO_2$ | $O_4$ |
|---|---|---|
| Fit window (nm) | 405 - 461 | 352 - 387 |
| Fitted absorption cross sections | $NO_2$ (298K, $I_0$ corrected) [1], $O_4$ (293K) [2], $NO_2$ (220K, $I_0$ corrected, pre-orthogonalized to $NO_2$ at 298K) [1], Ring, $H_2O$ [3], $O_3$ (223K) [4], $C_2H_2O_2$ [5] | $NO_2$(298K, $I_0$ corrected) [1], $O_4$ (293K) [2], HCHO (293K) [6], $O_3$ (223K) [4], BrO(223K) [7], Ring |
| Polynomial order | 5 | 5 |
| Intensity offset | Constant and first order | Constant and first order |
| Fraunhofer reference selection | Sequential | Sequential |

[1] (Vandaele et al., 1998), [2] (Thalman and Volkamer, 2013), [3] (Polyansky et al., 2018), [4] (Bogumil et al., 2003), [5] (Volkamer et al., 2005), [6] (Meller and Moortgat, 2000), [7] (Fleischmann et al., 2004)





**Table C2.** Summary of comparison of $dSCD$s at various elevation angles for the three different model set ups for the CM02 domain. The values in the bracket correspond to the RMSD, bias and R of the comparison of the geometric VCDs. The geometric $dAMF$ at $30°$ elevation angle equals 1 and hence the geometric VCD is the same as the measured $dSCD$.

| Telescope | elev. | MAX-DOAS Mean | TNO$_{fl}$ RMSD (%) | Bias (%) | $R$ | UBA$_{fl}$ RMSD (%) | Bias (%) | $R$ | UBA$_{di}$ RMSD (%) | Bias (%) | $R$ |
|---|---|---|---|---|---|---|---|---|---|---|---|
| T1 | 30 | 1.03E+16 | 60.54 (68.94) | -13.15 (-47.20) | 0.17 (0.16) | 82.30 (60.84) | 30.62 (-20.74) | 0.19 (0.17) | 99.81 (60.38) | 44.66 (-12.98) | 0.25 (0.27) |
| | 15 | 2.47E+16 | 59.58 | -24.77 | 0.19 | 68.12 | 12.79 | 0.19 | 72.80 | 23.25 | 0.31 |
| | 10 | 3.61E+16 | 61.05 | -30.15 | 0.23 | 64.82 | 4.45 | 0.21 | 64.24 | 13.40 | 0.34 |
| | 8 | 4.30E+16 | 62.51 | -32.60 | 0.26 | 64.22 | 0.42 | 0.23 | 61.36 | 8.69 | 0.36 |
| | 5 | 5.81E+16 | 64.29 | -36.96 | 0.38 | 62.07 | -7.45 | 0.34 | 56.60 | -0.39 | 0.45 |
| | 3 | 7.29E+16 | 66.52 | -39.54 | 0.50 | 61.40 | -12.50 | 0.46 | 55.41 | -5.96 | 0.54 |
| | 2 | 7.98E+16 | 65.99 | -38.96 | 0.55 | 61.36 | -11.88 | 0.52 | 54.36 | -5.28 | 0.60 |
| T2 | 30 | 1.05E+16 | 57.45 (69.22) | -13.65 (-47.10) | 0.29 (0.21) | 81.72 (60.60) | 30.53 (-20.51) | 0.30 (0.24) | 98.06 (57.14) | 44.78 (-13.10) | 0.39 (0.39) |
| | 15 | 2.52E+16 | 57.40 | -25.50 | 0.30 | 67.11 | 11.72 | 0.28 | 70.42 | 22.10 | 0.42 |
| | 10 | 3.74E+16 | 60.29 | -32.00 | 0.30 | 63.92 | 1.18 | 0.27 | 61.41 | 9.92 | 0.43 |
| | 8 | 4.48E+16 | 61.46 | -34.67 | 0.31 | 63.02 | -3.38 | 0.27 | 58.15 | 4.67 | 0.44 |
| | 5 | 6.07E+16 | 63.82 | -38.64 | 0.34 | 62.84 | -10.57 | 0.29 | 54.69 | -3.49 | 0.47 |
| | 3 | 7.76E+16 | 65.48 | -41.19 | 0.41 | 62.92 | -15.17 | 0.36 | 53.23 | -8.55 | 0.53 |
| | 2 | 8.72E+16 | 64.46 | -40.85 | 0.47 | 62.15 | -15.05 | 0.43 | 52.40 | -8.53 | 0.57 |
| T3 | 30 | 8.95E+15 | 62.48 (69.16) | -4.81 (-37.48) | 0.32 (0.26) | 90.92 (63.25) | 40.80 (-9.27) | 0.35 (0.32) | 113.73 (63.78) | 54.64 (-0.85) | 0.38 (0.39) |
| | 15 | 2.17E+16 | 57.55 | -16.03 | 0.32 | 70.84 | 21.89 | 0.36 | 81.04 | 33.58 | 0.40 |
| | 10 | 3.22E+16 | 59.07 | -22.09 | 0.30 | 65.31 | 11.78 | 0.35 | 69.79 | 21.77 | 0.40 |
| | 8 | 3.88E+16 | 59.98 | -24.96 | 0.30 | 63.16 | 6.84 | 0.35 | 65.32 | 16.06 | 0.41 |
| | 5 | 5.33E+16 | 61.32 | -29.00 | 0.33 | 60.74 | -0.78 | 0.38 | 60.06 | 7.20 | 0.43 |
| | 3 | 6.87E+16 | 60.41 | -30.92 | 0.43 | 58.16 | -5.38 | 0.47 | 56.52 | 1.47 | 0.49 |
| | 2 | 7.72E+16 | 57.21 | -29.23 | 0.54 | 56.34 | -4.59 | 0.57 | 55.82 | 1.34 | 0.54 |
| T4 | 30 | 8.91E+15 | 62.59 (66.03) | -6.27 (-37.47) | 0.24 (0.26) | 85.33 (61.05) | 36.39 (-9.80) | 0.29 (0.28) | 104.84 (66.91) | 50.58 (-0.44) | 0.27 (0.27) |
| | 15 | 2.20E+16 | 57.89 | -19.17 | 0.30 | 65.91 | 15.90 | 0.34 | 74.38 | 27.05 | 0.37 |
| | 10 | 3.20E+16 | 59.54 | -24.24 | 0.32 | 62.37 | 7.74 | 0.36 | 66.56 | 17.66 | 0.40 |
| | 8 | 3.79E+16 | 60.49 | -26.15 | 0.34 | 61.36 | 4.56 | 0.38 | 63.95 | 13.99 | 0.42 |
| | 5 | 5.09E+16 | 62.51 | -29.25 | 0.39 | 60.41 | -0.79 | 0.44 | 60.63 | 7.77 | 0.46 |
| | 3 | 6.27E+16 | 59.74 | -29.22 | 0.50 | 56.92 | -1.98 | 0.54 | 58.47 | 5.53 | 0.52 |
| | 2 | 6.62E+16 | 57.55 | -26.12 | 0.50 | 57.55 | 0.59 | 0.54 | 61.24 | 8.02 | 0.47 |



**Table C3.** Same as Table C2, but for cloud-free cases only

| Telescope | elev. | MAX-DOAS Mean | TNO$_{fl}$ RMSD (%) | Bias (%) | $R$ | UBA$_{fl}$ RMSD (%) | Bias (%) | $R$ | UBA$_{di}$ RMSD (%) | Bias (%) | $R$ |
|---|---|---|---|---|---|---|---|---|---|---|---|
| T1 | 30 | 9.99E+15 | 53.11 (64.36) | -15.61 (-47.26) | 0.27 (0.22) | 81.00 (59.81) | 27.62 (-20.72) | 0.19 (0.14) | 77.95 (49.73) | 33.94 (-18.11) | 0.36 (0.37) |
| | 15 | 2.48E+16 | 54.68 | -25.35 | 0.25 | 69.78 | 12.31 | 0.14 | 57.68 | 16.16 | 0.38 |
| | 10 | 3.74E+16 | 56.33 | -30.71 | 0.28 | 65.59 | 3.96 | 0.15 | 49.96 | 6.77 | 0.42 |
| | 8 | 4.52E+16 | 57.56 | -33.14 | 0.31 | 63.99 | -0.05 | 0.19 | 46.84 | 2.26 | 0.46 |
| | 5 | 6.39E+16 | 59.76 | -38.27 | 0.43 | 60.09 | -9.3 | 0.31 | 42.16 | -8.02 | 0.59 |
| | 3 | 8.26E+16 | 61.72 | -41.33 | 0.55 | 57.49 | -15.24 | 0.48 | 40.7 | -14.44 | 0.72 |
| | 2 | 9.18E+16 | 61.46 | -41.01 | 0.61 | 57.13 | -14.93 | 0.54 | 38.47 | -14.33 | 0.78 |
| T2 | 30 | 1.00E+16 | 54.01 (67.18) | -16.49 (-46.33) | 0.33 (0.22) | 81.72 (62.44) | 26.81 (-19.24) | 0.29 (0.20) | 78.27 (50.64) | 35.09 (-15.59) | 0.50 (0.46) |
| | 15 | 2.49E+16 | 57.55 | -25.62 | 0.3 | 72.69 | 11.6 | 0.24 | 59.77 | 16.88 | 0.49 |
| | 10 | 3.80E+16 | 60.75 | -31.67 | 0.29 | 69.54 | 1.52 | 0.22 | 53.45 | 5.59 | 0.49 |
| | 8 | 4.61E+16 | 61.93 | -34.16 | 0.29 | 68.37 | -2.89 | 0.22 | 51.36 | 0.64 | 0.5 |
| | 5 | 6.43E+16 | 64.84 | -37.9 | 0.31 | 68.08 | -9.93 | 0.24 | 50.08 | -7.22 | 0.52 |
| | 3 | 8.44E+16 | 66.28 | -40.48 | 0.39 | 67.36 | -14.62 | 0.31 | 49.11 | -12.29 | 0.58 |
| | 2 | 9.61E+16 | 64.61 | -40.05 | 0.46 | 65.74 | -14.35 | 0.39 | 47.08 | -12.4 | 0.64 |
| T3 | 30 | 8.91E+15 | 58.00 (68.07) | -9.76 (-37.79) | 0.40 (0.28) | 82.67 (62.54) | 34.17 (-9.08) | 0.45 (0.37) | 84.91 (53.39) | 41.66 (-6.11) | 0.55 (0.52) |
| | 15 | 2.24E+16 | 55.14 | -18.98 | 0.36 | 68.32 | 18.32 | 0.42 | 61.5 | 22.44 | 0.55 |
| | 10 | 3.38E+16 | 56.52 | -24.13 | 0.33 | 63.71 | 9.46 | 0.39 | 53.7 | 12.4 | 0.54 |
| | 8 | 4.13E+16 | 57.14 | -26.89 | 0.33 | 61.36 | 4.62 | 0.39 | 50.55 | 7.06 | 0.53 |
| | 5 | 5.82E+16 | 57.95 | -30.25 | 0.36 | 58.75 | -2.26 | 0.41 | 47.76 | -0.61 | 0.54 |
| | 3 | 7.75E+16 | 57.8 | -32.31 | 0.45 | 56.55 | -7.26 | 0.49 | 45.44 | -6.37 | 0.6 |
| | 2 | 8.85E+16 | 53.98 | -30.14 | 0.58 | 53.96 | -6.01 | 0.61 | 42.91 | -6.14 | 0.69 |
| T4 | 30 | 8.87E+15 | 54.53 (61.54) | -11.96 (-39.70) | 0.36 (0.35) | 78.55 (56.61) | 30.37 (-11.68) | 0.39 (0.36) | 75.34 (51.84) | 35.57 (-8.34) | 0.47 (0.43) |
| | 15 | 2.23E+16 | 52.44 | -21.76 | 0.39 | 63.84 | 14.15 | 0.4 | 54.82 | 17.77 | 0.55 |
| | 10 | 3.33E+16 | 54.88 | -26.7 | 0.4 | 60.3 | 6.04 | 0.41 | 48.91 | 8.73 | 0.56 |
| | 8 | 4.01E+16 | 55.78 | -28.73 | 0.41 | 58.76 | 2.6 | 0.43 | 46.69 | 4.91 | 0.58 |
| | 5 | 5.57E+16 | 56.96 | -31.89 | 0.47 | 56.21 | -3.01 | 0.48 | 43.61 | -1.4 | 0.63 |
| | 3 | 7.33E+16 | 57.69 | -34.05 | 0.55 | 53.86 | -7.31 | 0.57 | 41.87 | -6.38 | 0.69 |
| | 2 | 7.69E+16 | 54.18 | -30.67 | 0.57 | 53.53 | -4.4 | 0.58 | 41.32 | -4.62 | 0.67 |



*Author contributions.* VK and TW prepared the manuscript with inputs from all the co-authors. JR performed the MAX-DOAS measurements, $dSCD$s and $dbAMF$ retrieval with inputs from SB and TW. AK, MM and BS helped VK with the model set up and simulations. VK and AP pre-processed the TNO and UBA emission data so as to render it suitable for MECO(n). VK performed the cloud classification and data analyses related to model evaluation.

*Competing interests.* Authors declare no competing interests.

*Acknowledgements.* Vinod Kumar acknowledges the Max Planck Society and Alexander von Humboldt foundation for providing financial support in the form of postdoctoral fellowship. We acknowledge the Umweltbundesamt and Deutscher Wetterdienst for in situ measurement data and UBA emission inventory. The model simulations were performed at the supercomputer MISTRAL of German Climate Computing Centre. We thank Patrick Jöckel for maintenance and managing the license of the MESSy code. We thank Jeroen J.P. Kuenen for providing the TNO MACC III anthropogenic emission data and ECWMF for ERA-INTERIM reanalysis data used for nudging EMAC. Data analysis and visualization was performed using python 3.8 and standard libraries including numpy, scipy, pandas, netCDF4, h5py, matplotlib, cartopy and geopy.



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
