# Peer review of "Evaluation of the coupled high-resolution atmospheric chemistry model system MECO(n) using in situ and MAX-DOAS $NO_2$ measurements"

_Atmospheric Measurement Techniques, 2021_

## Author Comment (AC1)

This is a nice piece of work, I enjoy reading it. This work provide a comprehensive evaluation of NOx high-resolution simulation over Germany. NOx is one of the most important tracer gases in the atmosphere, which largely impacts photochemistry and generate secondary pollutants, including ozone and nitrate. High resolution simulation is critical to improve the non-linear photochemical processes and heterogeneous processes, the latter of which is also important for both air quality and climate, eg. (Chen et al., 2020). The work uses a broad surface observation network and DOAS column observations of NOx to provide a valuable evaluation of the high-resolution results using different emission inventories, and help us understand the uncertainties in model and inventories. I am more familiar with modelling and possibly not the best person to comment on the observation part, while, I have collaborated with MPIC a few years ago, I trust the data is of high quality from their rigorous research style. And authors have described the observation method in details and it looks convincing for me. This manuscript is well organized and written. I am happy to recommend it for publishing.

Thanks a lot for the encouraging remarks and highlighting the importance of this work. We have noted the relevance of high resolution modelling for non-linear photochemical processes and heterogeneous processes and accordingly modified lines 20-22 of the original manuscript as follows:

*"The high spatial resolution of these models allows us to resolve localized emissions (e.g., industrial and urban clusters), quantify their impacts on non-linear photochemical processes, e.g., ozone production (Vinken et al., 2014; Visser et al., 2019; Mertens et al., 2020) as well as on heterogeneous processes e.g., particulate nitrate production (Chen et al., 2020)."*

A few minor comments may help improve the discussion.

We have undertaken the revision suggested by this reviewer. The questions and suggestions are marked in blue color while our responses are shown in black color.

1) line 39, "secondary chemistry", may be better using secondary pollution?

Done.

2) A little more information on the emission inventory would be helpful. Such as, the resolution of TNO and UBA inventory, and how does seasonal variation considered? What does "fl" subtitle mean, better introduce it.

Thanks a lot for the suggestion. We have added the following text in lines 131-132 of the revised manuscript to provide information about the spatial resolution of TNO and UBA inventory

*"The spatial resolution of TNO MACC III and UBA emissions are 0.0625° (latitude) × 0.125° (longitude) and 1km × 1km, respectively."*

In appendix A (line 558-560) of the original manuscript, we provide the information about how we considered the monthly variability.

*"Both TNO MACC III and UBA emissions are available at a temporal resolution of one year. The monthly profiles of the anthropogenic emissions depend on the emitted species, emissions sectors and the country. For Europe, these factors are also provided by Builtjes et al. (2002), and, are used to create monthly resolution emissions."*

We have modified lines 132-134 of the original manuscript as follows to introduce the "*fl*" and "*di*" subscripts:

*The subscript "di" in Table 1 indicates the use of the diurnal and day-of-the-week variability in NO$_x$ and CO emissions from the road transport and residential and non-industrial combustion sectors (see appendix A for further details). Similarly, the subscript "fl" (e.g. in the TNO$_{fl}$ and UBA$_{fl}$ set ups)*

*indicates that constant anthropogenic emissions (and a "flat" diurnal pattern) are used for the complete month.*

3) In conclusion, up to 50% higher human emission of NOx only have a minor effect on ambient VMRs and dSCDs, and authors think this possibly stem from the short lifetime of NOx. I feel we may want a more careful discussion here, because, no matter what time scale it is for NOx lifetime, NOx concentration is a result of the equilibrium between sources and sinks. And 50% higher of anthropogenic source has a minor impact on surface concentration, I feel it could due to two reasons: 1) anthropogenic emission is a minor source in afternoon, clearly this cannot be true over Germany, or 2) boundary layer vertical mixing is high in afternoon, this maybe more likely to be the reason. You may want to take a look of the column NOx value, VCDs. If there is a clear increase of VCDs, it could be an evidence of vertical mixing.

Thanks a lot for the feedback and the suggestion. Indeed the boundary layer height plays an important role in governing the surface volume mixing ratios (VMR) of $NO_2$, as indicated in lines 489-491 of the original manuscript:

*"$NO_2$ has a short lifetime of a few hours in the daytime, which together with a strong mixing in the daytime boundary layer compensates even for $\sim$ 50% higher emissions (an increase in emissions from the transport sector by a factor of two would translate to an overall increase of $\sim$ 50% in $NO_x$ emissions)"*

We agree with the reviewer that, during the daytime, the minor effect of increased emissions on ambient VMRs and VCDs is not entirely due to the short lifetime of $NO_2$. We also feel that the use of the word "compensate" is not appropriate here, and we correct this in the revised manuscript.

Furthermore, as suggested by the reviewer, in order to disentangle the effect of vertical mixing in the daytime boundary layer from that of photochemistry, we compare the mean diurnal patterns of both the tropospheric VCD and surface VMR of $NO_2$ at the background site Mainz Mombach for the $UBA_{fl}$ simulations. This set up was chosen because it does not consider any diurnal modulation in emissions and observed diurnal patterns are due to chemical and meteorological effects. We observe that the diurnal variability is much stronger for the surface VMRs (363% peak to peak) as compared to tropospheric VCDs (83 % peak to peak).

This further confirms the stronger impact of boundary layer evolution as compared to the lifetime on the surface mixing ratios of $NO_2$.

[Figure]

*Figure 1: Mean hourly diurnal profiles of simulated NO₂ tropospheric VCDs (red) and surface volume mixing ratios (blue) at a background site Mainz Mombach (8.02 °E, 50.02 °N) for the model set up UBA_fl (diurnal variability of road transport and non-industrial combustion sectors not considered). The solid line and markers represent the mean while the shaded region shows the inter-quartile variability range.*

Considering these points, we have added Figure 1 shown above in the appendix of the revised manuscript and modified lines 489-492 of the original manuscript as follows:

*"Concerning the diurnal variation, both UBA_fl and UBA_di show smaller NO₂ dSCDs and surface VMRs during the daytime which agrees reasonably well with the measurements in both set ups. However, stronger discrepancies are observed during early morning and late evening for UBA_fl. The discrepancy is even larger for the UBA_fl surface concentration at night-time (Figure 13). While emissions primarily drive the surface VMRs during night-time, dilution in the higher boundary layer and chemical loss due to OH in the daytime counters the stronger emissions. We further investigated the diurnal patterns of VCDs and surface VMRs in the UBA_fl set-up (Figure C4), and found that the magnitude of diurnal modulation in the hourly mean was ~350% peak to peak, while the same was ~80% for VCDs. This difference in the magnitude of diurnal variability indicates a stronger role of the boundary layer height evolution as compared to the chemical loss due to short lifetime towards off-setting the effect of higher NOₓ emissions during the daytime."*

This has also been updated in the abstract (lines 16-18 of the original manuscript) and conclusion (line 537-538 of the original manuscript).

*"Accounting for diurnal and daily variability in the monthly resolved anthropogenic emissions was found to be crucial for the accurate representation of time series of measured NO₂ VMR and dSCDs and is particularly critical when vertical mixing is suppressed and the atmospheric lifetime of NO₂ is relatively long."*

*"For the afternoon hours, however, even up to 50% higher anthropogenic NOₓ emissions only have a minor effect on ambient VMRs and dSCDs due to enhanced dilution in the high daytime planetary boundary layer and its short atmospheric lifetime."*

References:

Chen, Y., Cheng, Y., Ma, N., Wei, C., Ran, L., Wolke, R., Größ, J., Wang, Q., Pozzer, A., Denier van der Gon, H. A. C., Spindler, G., Lelieveld, J., Tegen, I., Su, H., and Wiedensohler, A.: Natural sea-salt

emissions moderate the climate forcing of anthropogenic nitrate, Atmos. Chem. Phys., 20, 771-786, 10.5194/acp-20-771-2020, 2020.

---

## Author Comment (AC2)

Kumar et al. present a confrontation of $NO_2$ simulations performed with a high-resolution atmospheric chemistry model, set up over southwest Germany, with surface mixing ratio observations and ground-based MAX-DOAS observations of $NO_2$ vertical column densities and differential slant column densities. The simulations are performed using two emission inventories with different spatial resolutions and emission totals. An additional simulation is performed where sector-specific temporal emission profiles are applied for the high-resolution inventory. The authors report the best model performance using the high-resolution inventory with temporal variation in emissions. Next, the authors derive differential slant column densities from the model simulations and compare these to MAX-DOAS observations under different viewing angles. This allows an evaluation of the spatial and vertical distribution of $NO_2$ in the vicinity of the MAX-DOAS instrument.

The evaluation of the atmospheric chemistry model is comprehensive and detailed. The evaluation of dSCDs to evaluate the spatial and vertical distribution of $NO_2$ in the lower atmosphere seems innovative and reproducible in other model evaluation studies (from my atmospheric chemistry modelling perspective). The overall quality of the presented figures and tables, as well as their discussion in the text, is of good quality. Therefore, the manuscript is suitable for publication in AMT after addressing the following questions.

Thanks a lot for assessing our study and the constructive comments. In this document, we address the individual comments and questions of this reviewer. Our responses are presented in black color while the questions are presented in blue color. The revisions in the manuscript are shown in *italics*.

- The difference between the TNO-MACC-III and UBA anthropogenic emission inventories (70%) is remarkable and deserves further discussion.
    1. Can anything be concluded regarding the agreement per source sector?

Thanks a lot for highlighting this point. Here we admit that there was a mistake related to the selection of country mask in the calculation of total emissions from the TNO-MACC-III. In the original manuscript, the total $NO_x$ emissions from the TNO-MACC-III inventory within Germany was 214 Gg(N) for 2011, while the correct value is 288 Gg(N). We have corrected it at lines 128 and 276-280 of the original manuscript. The actual difference in the $NO_x$ emissions between the two $NO_x$ emission inventories is 21% with respect to UBA emissions (27% with respect to TNO MACC III emissions).

We also show the sector-wise difference in figure 1 below. The individual sectors are Agriculture (AGR), Energy industries (ENE), Other Industries (IND), Residential and non-industrial combustion (RCO), Fossil fuel production and distribution (REF), Water navigation (SHP), Solvent and other product use (SOL), Road transport (TRA) and Waste collection, treatment and disposal activities (WST). TNO MACC III deviates most from the UBA dataset in the road transport, energy industries and other industries sectors towards lower estimates, while ship emissions are higher in TNO MACC III. It should also be noted that agricultural emissions are not included in TNO MACC III inventory (Kuenen et al., 2014), while these are included in UBA inventory. Here we would like to point out that in our simulations using UBA emission inventory, emissions due to the use of organic and inorganic fertilizers are doubly counted within Germany, as these are also calculated online using the ONEMIS submodel. We have checked the contribution of these two sources in the total $NO_x$ emissions within Germany, and these are 8% of the total. However, this is within the uncertainty of the $NO_x$ emissions in the inventories, which are in the order of 20% (Solazzo et al., 2021).

[Figure]

*Figure 1 Contribution of individual sectors towards the total NO$_x$ emissions within Germany for TNO MACC III emission inventory for 2011 (blue) and UBA emission inventory for 2018 (orange).*

This information and Fig. 1 are now included in Appendix A of the revised manuscript.

2.  How do EDGAR emission totals compare to both inventories (as an independent estimate)?

    We have calculated the EDGAR version 5.0AP (https://edgar.jrc.ec.europa.eu/dataset_ap50) total NO$_x$ emissions for years 2011 and 2015 for Germany. These are 389.2 and 366.9 Gg(N) per year respectively. We have also added this information in the revised manuscript (line 161).

3.  I believe TNO-MACC-III distinguishes between gridded sources (which have unit mass per grid cell per year) and point sources (which have unit mass per year). The latter should be added after interpolation to the destination model grid in order to conserve the mass balance. Not accounting for this leads to inaccurate representations of local emission peaks, and may affect domain emission totals. Please further discuss the strategy to interpolate emission data.

Thanks a lot for highlighting this issue. We apprehended this issue and have carefully checked the TNO MACC III emission inventory. There point sources are distinguished during the preparation stage and are subsequently distributed in the grid as suggested by the reviewer. In this context, we would like to cite the excerpt from the TNO MACC emissions inventory description paper (Kuenen et al., 2014).

"*Point sources were spatially distributed using the specific location of the point source*"

4.  It would be good to embed this finding in the context of the literature: e.g. Travis et al. (2016) suggest sector-specific emission reductions of 30-60% over the Southeast US, and Visser et al. (2019) report European satellite-derived emission totals of ±50% higher compared to TNO-MACC.

Thanks for the suggestions. We have modified lines 282-285 of the original manuscript as follows to include the additional discussion:

*"Top down emission estimates using OMI NO$_2$ measurement also indicated an underestimation of NO$_x$ emission by more than 50% over western Germany and other parts of Europe by the TNO MACC III emission inventory (Visser et al., 2019). It should be noted that the exclusion of soil NO$_x$ emissions from the TNO MACC III inventory also contributed towards the large underestimation in the estimates by Visser et al. (2019). The underestimation of the a priori NO$_x$ emissions is the most important factor for the large negative bias in the TNO$_{fl}$ set-up. Getting up to date emission inventories is difficult, and we could only get this data for Germany, but for future studies with simulation involving larger domains, which include more countries, it is recommended to use more up-to-date emission inventories. The situation of underestimation of NO$_x$ emissions over Germany (and most parts of Europe) is different from that observed in the USA, where national emissions inventories are biased high by as much as 30-60% (Travis et al., 2016)"*

- Based on the model comparison with MAX-DOAS observations, can anything be concluded regarding the representativeness of surface emissions from different sources in relation to the model-observation agreement (e.g. T2 in direction of Mainz (anthropogenic footprint), vs. T4 in direction of agricultural areas).

Thanks for the question. In lines 437-439 of the original manuscript, we mention that we did not observe a large difference in the measured and simulated dSCDs along the four viewing directions. One of the plausible explanation of this observation could be the prevailing wind directions (panel (a) of Figure 6), as for most of the time, easterly winds bring the air mass from the urban locations to the agricultural lands.

However, we have evaluated the spatial patterns of the NO$_x$ emissions from the UBA inventory for the individual sectors classified as Agriculture (AGR), Energy industries (ENE), Other Industries (IND), Residential and non-industrial combustion (RCO), Fossil fuel production and distribution (REF), Water navigation (SHP), Solvent and other product use (SOL), Road transport (TRA) and Waste collection, treatment and disposal activities (WST). These spatial patterns are shown in the figure below and the four viewing directions of the MAX-DOAS instrument are overlaid on the maps. While it is difficult to estimate the accuracy of the emissions quantitatively, qualitatively, we can infer that the emissions sources are represented reasonably. For example, while anthropogenic emissions sources (e.g. RCO, IND and ENE) are concentrated towards T2 and T3, agricultural emissions sources are more abundant along T4 and west of T3. Ship emissions are along the Rhine river while the road transport emissions follow the motorways as well as are concentrated in the urban regions.

[Figure]

*Figure 2 Spatial patterns of NOₓ emissions from the UBA inventory for 2018 around the MAX-DOAS measurement location. The viewing directors of the four telescopes are overlaid on top of the emissions maps. Broadly, the telescopes T2 and T3 point towards the city centre and urban areas, T4 points towards the agricultural lands and T1 points towards anthropogenic sources (in the near vicinity) and small forests.*

- Appendix B: this is a highly relevant discussion, and most (if not all) models struggle to accurately capture the diurnal cycle in $O_3$. I believe this section can be strengthened by pointing this out, for example by referring to regional model intercomparison efforts with similar results (e.g. Solazzo et al. 2012; Im et al., 2015). Why are $O_3$ simulations only moderately sensitive to substantial $NO_x$ emission differences? Can you detect an effect of model resolution on $O_3$ mixing ratios (e.g. by comparing domains CM07 and CM02)?

Thanks a lot for indicating the importance of this section and the relevant discussion. We have extended the discussion in appendix B by adding the following lines to the manuscript (lines 622-628):

*"Over the European region, regional models usually struggle to capture the diurnal variability of surface $O_3$ VMRs as indicated by AQMEII (Air Quality Model Evaluation International Initiative) studies (Solazzo et al., 2012; Im et al., 2015). For example, in WRF-CHEM, the night-time overestimation of surface $O_3$ was found to be due to an underestimation of $O_3$ titration by $NO_x$, while that in COSMO-MUSCAT was due to relatively weaker dry deposition fluxes and inaccurate representation of vertical mixing. A night-time overestimation over central Europe was also observed*

*for global model EMAC, though the bias was smaller as compared to MECO(n) (Mertens et al., 2016)."*

[Figure]

*Figure 3 Top) Spatial patterns of monthly mean O₃ volume mixing ratios for the UBA_{di} set up for CM02 domain (left panel) and CM07 domain (right panel). The corresponding monthly mean measured O₃ VMRs are shown as markers overlaid on the map. The numbers in the map show major cities similar to that in Figure 4 of the original manuscript. The bottom panel shows the linear regression between the measured and simulated monthly mean O₃ VMRs at various measurement stations.*

For the UBA_{di} setup, we compared the spatial patterns and mean ozone surface volume mixing ratios (VMR) at various measurement stations in the CM07 and CM02 domain (Figure 3). From the spatial patterns, we could conclude that the fine resolution model (left panel) simulates the spatial patterns in greater detail. Its improved spatial variability is obvious from the scatter plot, where we could find better accountability (slope) corresponding to the measured values. At coarser resolution, we observe moderately higher (~2ppb) domain mean ozone mixing ratio (Figure 4). This further supports our hypothesis of a $NO_x$ saturated ozone production regime (lines 600-601 of the original manuscript); as the $NO_2$ mixing ratios are lower by ~0.2 ppb in the 7 km resolution domain as compared to the 2.2 km

resolution domain.

[Figure]

*Figure 4 Spatial patterns of difference in simulated monthly mean O₃ surface VMRs (left panel) and mean NO₂ surface VMRs (right panel) between CM07 and CM02 domains of the UBA$_{di}$ set up.*

In mid-Europe, the non-anthropogenic sources (e.g. biogenic, methane oxidation, stratospheric intrusion) contribute to > 55% of the surface ozone production in the summertime (Mertens et al., 2020). The contribution is even higher for the CM02 domain region presented in the study. The stronger difference in $NO_x$ emissions between the TNO and UBA emissions are with respect to the road transport emissions, energy industries and other industries (see Figure 1). The contribution of road transport emissions towards surface ozone in the summertime are 12-14%, while that for all other anthropogenic sources, including energy industries is >30%.

Hence, we believe that a relatively weaker contribution of anthropogenic emissions as well as a $NO_x$ saturated ozone production regimes are the reason behind the moderate sensitivity of surface ozone towards $NO_x$ emissions in our study.

We have added the following lines in the revised manuscript to discuss the above-mentioned points (lines 639-642 of the revised manuscript):

*"It is interesting to note that an increase of ~27% anthropogenic $NO_x$ emissions by using the UBA emission inventory shows little change in surface O₃ VMRs. In addition to the $NO_x$ saturated ozone production regime, a relatively weaker contribution of anthropogenic emissions towards the ozone production in summertime (Mertens et al., 2020b) could also be a plausible reason for this weak effect."*

Specific comments

- Line 96: change '6 hourly' to '6-hourly'

Done

- Figure 1: Please increase the font size of the annotated text in the zoomed panel for increased legibility

Done. The new figure is also shown below.

[Figure]

- Line 151-154: I suggest to give some more context to the soil NO$_x$ emission totals and how they compare to other estimates. This is especially relevant in the context of the increasing importance of soil NO$_x$ due to decreasing anthropogenic NO$_x$ sources (see e.g. Skiba et al. 2020)

Thanks a lot for suggesting this point. The soil NO$_x$ emissions are highly variable as these are dependent on various meteorological factors including ambient temperature and rainfall. Since we have performed model simulations only for one month (May 2018), it is difficult to accurately compare with other estimates, which provide annual soil NO$_x$ emissions estimates. However, we attempt to do so by considering the monthly temporal profiles of agricultural soil NO$_x$ emissions prescribed by Crippa et al. (2020). Upon application of the monthly factors, the 5.9 Gg(N) soil NO$_x$ emissions over complete Germany in the CM07 domain scales to ~33.7 Gg(N) annually. For reference, the soil NO$_x$ emissions combined due to the application of inorganic nitrogen fertilizers and animal manure was 30.1 Gg(N) for the year 2018, according to the UBA emission inventory. However, for EDGAR version 5.0, the total soil NO$_x$ emissions directly from agricultural soils and Manure management combined for May 2015 for Germany were 19 Gg(N).

In the revised manuscript, we have added the comparison with respect to the EDGAR version 5.0 soil NO$_x$ emissions in lines 160-161:

*"The soil NO$_x$ emission calculated online are smaller as compared to the estimated 19.0 Gg(N) for May 2015 from agricultural soils and Manure management sectors combined by EDGAR v5.0 (Crippa et al, 2020)"*

- Line 370-371: The reference to Figure C1 now refers to the cloud classification figure, and I cannot find the figure showing Pi-MAX VCDs elsewhere in the text. Please include this figure.

Thanks for pointing this out. This was a mistake in figure labels. The π-MAX VCD are shown as black markers in Figure 6 of the original manuscript. Accordingly, we have corrected the reference in the text as well.

- Line 589: change 'deposited' to 'deposition'

Done

References

Im, U., R. Bianconi, E. Solazzo, I. Kioutsioukis, A. Badia, A. Balzarini, R. Baró et al.: Evaluation of operational on-line-coupled regional air quality models over Europe and North America in the context of AQMEII phase 2. Part I: Ozone. Atmos. Environ., 115, 404-420, https://doi.org/10.1016/j.atmosenv.2014.09.042, 2015.

Skiba, U., Medinets, S., Cardenas, L.M., Carnell, E.J., Hutchings, N.J. Amon, B.: Assessing the contribution of soil $NO_x$ emissions to European atmospheric pollution, Environ. Res. Lett, 16 (025009), https://doi.org/10.1088/1748-9326/abd2f2, 2020.

Solazzo, E., R. Bianconi, R. Vautard, K. Wyat Appel, M.D. Moran, C. Hogrefe, B. Bessagnet et al.: Model evaluation and ensemble modelling of surface-level ozone in Europe and North America in the context of AQMEII. Atmos. Environ., 53, 60-74, https://doi.org/10.1016/j.atmosenv.2012.01.003, 2012.

Travis, K. R., Jacob, D. J., Fisher, J. A., Kim, P. S., Marais, E. A., Zhu, L., Yu, K., Miller, C. C., Yantosca, R. M., Sulprizio, M. P., Thompson, A. M., Wennberg, P. O., Crounse, J. D., St. Clair, J. M., Cohen, R. C., Laughner, J. L., Dibb, J. E., Hall, S. R., Ullmann, K., Wolfe, G. M., Pollack, I. B., Peischl, J., Neuman, J. A., and Zhou, X.: Why do models overestimate surface ozone in the Southeast United States?, Atmos. Chem. Phys., 16, 13561–13577, https://doi.org/10.5194/acp-16-13561-2016, 2016.

Visser, A. J., Boersma, K. F., Ganzeveld, L. N., and Krol, M. C.: European $NO_x$ emissions in WRF-Chem derived from OMI: impacts on summertime surface ozone, Atmos. Chem. Phys., 19, 11821–11841, https://doi.org/10.5194/acp-19-11821-2019, 2019.

Crippa, M., Solazzo, E., Huang, G., Guizzardi, D., Koffi, E., Muntean, M., Schieberle, C., Friedrich, R., and Janssens-Maenhout, G.: High resolution temporal profiles in the Emissions Database for Global Atmospheric Research, Scientific Data, 7, 121, 10.1038/s41597-020-0462-2, 2020.
Kuenen, J. J. P., Visschedijk, A. J. H., Jozwicka, M., and Denier van der Gon, H. A. C.: TNO-MACC_II emission inventory; a multi-year (2003-2009) consistent high-resolution European emission inventory for air quality modelling, Atmos. Chem. Phys., 14, 10963-10976, 10.5194/acp-14-10963-2014, 2014.
Mertens, M., Kerkweg, A., Grewe, V., Jöckel, P., and Sausen, R.: Attributing ozone and its precursors to land transport emissions in Europe and Germany, Atmos. Chem. Phys., 20, 7843-7873, 10.5194/acp-20-7843-2020, 2020.
Solazzo, E., Crippa, M., Guizzardi, D., Muntean, M., Choulga, M., and Janssens-Maenhout, G.: Uncertainties in the Emissions Database for Global Atmospheric Research (EDGAR) emission inventory of greenhouse gases, Atmos. Chem. Phys., 21, 5655-5683, 10.5194/acp-21-5655-2021, 2021.